# A nucleotide-sensing oligomerization mechanism that controls NrdR-dependent transcription of ribonucleotide reductases

Inna Rozman Grinberg [1,4], Markel Martínez-Carranza [1,2,4], Ornella Bimai[1], Ghada Nouaïria[1], Saher Shahid[1], Daniel Lundin[1], Derek T. Logan [3✉], Britt-Marie Sjöberg [1✉] & Pål Stenmark [1,2✉]

Ribonucleotide reductase (RNR) is an essential enzyme that catalyzes the synthesis of DNA building blocks in virtually all living cells. NrdR, an RNR-specific repressor, controls the transcription of RNR genes and, often, its own, in most bacteria and some archaea. NrdR senses the concentration of nucleotides through its ATP-cone, an evolutionarily mobile domain that also regulates the enzymatic activity of many RNRs, while a Zn-ribbon domain mediates binding to NrdR boxes upstream of and overlapping the transcription start site of RNR genes. Here, we combine biochemical and cryo-EM studies of NrdR from Streptomyces coelicolor to show, at atomic resolution, how NrdR binds to DNA. The suggested mechanism involves an initial dodecamer loaded with two ATP molecules that cannot bind to DNA. When dATP concentrations increase, an octamer forms that is loaded with one molecule each of dATP and ATP per monomer. A tetramer derived from this octamer then binds to DNA and represses transcription of RNR. In many bacteria — including well-known pathogens such as Mycobacterium tuberculosis — NrdR simultaneously controls multiple RNRs and hence DNA synthesis, making it an excellent target for novel antibiotics development.

[1] Department of Biochemistry and Biophysics, Stockholm University, SE-10691 Stockholm, Sweden. [2] Department of Experimental Medical Science, Lund University, Box 118, SE-22100 Lund, Sweden. [3] Biochemistry and Structural Biology, Department of Chemistry, Lund University, SE-22100 Lund, Sweden. [4]These authors contributed equally: Inna Rozman Grinberg, Markel Martínez-Carranza. ✉email: derek.logan@biochemistry.lu.se; britt-marie.sjoberg@dbb.su.se; pal.stenmark@dbb.su.se

Ribonucleotide reductase is the only known de novo path to produce the deoxyribonucleotides needed for DNA synthesis and is, therefore, a promising drug target[1,2]. Three homologous classes, differing in their radical-generating cofactor and oxygen requirements, have been identified[3,4]. Class I RNR, encoded by *nrdA* (catalytic subunit) and *nrdB* (radical-generating subunit), is oxygen-requiring, class II RNR, encoded by *nrdJ*, is adenosylcobalamin-dependent, and class III RNR, encoded by *nrdD*, is oxygen-sensitive. Transcription of RNR genes and, in many cases, the activity of the enzyme need to be controlled to ensure correct concentrations of dNTPs to maintain genetic accuracy during replication and repair[5,6]. Moreover, in genomes encoding more than one RNR, the best-suited RNR needs to be selected to fit the cell cycle phase or environmental conditions at each time.

The RNR-specific transcriptional repressor NrdR regulates the expression of RNR genes in most bacteria but is rare in archaea and lacking in eukaryotes[7]. NrdR is found in genomes encoding all possible combinations of RNR classes, including species that have a single RNR operon, and is found in pathogens causing antibiotic resistance problems such as *Mycobacterium tuberculosis*, *Pseudomonas aeruginosa*, and *Staphylococcus aureus*. NrdR binds to palindromic "NrdR boxes" upstream of RNR operons that often overlap the RNA polymerase binding site[7–10]. NrdR studies have primarily been focused on *Streptomyces coelicolor* with class I and II operons[7,11], *Escherichia coli* with two class I and one class III operons[9,12], and *P. aeruginosa* with one operon of each RNR class[10]. Several studies have shown that deletion of the *nrdR* gene results in a general increase in expression of RNR[9,10,13–15].

Several hypotheses on how NrdR regulates RNR expression have been proposed[10–12,16], but its structure and mechanism of action have remained elusive. Here, for the first time, we combine biochemical and structural studies to elucidate which effectors are required for binding of *S. coelicolor* NrdR to DNA and solve the structure of NrdR loaded with nucleotides bound to DNA and in two unbound forms. The cryo-EM structures are unprecedented, revealing a widespread, structurally and functionally novel mechanism of transcriptional regulation. Deep functional and evolutionary relationships are revealed between this regulator of RNR genes and the RNR enzyme itself, since the nucleotide-binding domain of NrdR, the ATP-cone, also allosterically regulates the enzymatic activity of RNR[17].

## Results and discussion

**Structure of NrdR**. NrdRs (140–220 residues) consist of an N-terminal Zn-ribbon domain followed by an ATP-cone domain and in some cases a C-terminal part of variable length and unknown function (Fig. 1a and Supplementary Fig. 1)[7,18]. The Zn-ribbon domain is approximately 43 residues long, harbors two conserved CxxC motifs that coordinate a zinc ion, and binds to the highly conserved NrdR boxes. Interestingly, our results reveal that the ATP-cone domain of NrdR binds nucleotides differently to the ATP-cones of RNRs, the enzymes whose transcription NrdR regulates. The ATP-cone of NrdR constitutes a new class that has two nucleotide-binding sites and the ability to bind ATP and dATP simultaneously. In RNR the ATP-cone promotes a variety of protein–protein interactions resulting in different active or inactive oligomeric states[19–26]. Here we show that the ATP-cone in NrdR also controls oligomerization, in this case, to regulate DNA binding.

**DNA binding requires both dATP and ATP**. In microscale thermophoresis (MST) experiments, neither ATP alone nor dATP alone promoted the binding of NrdR to the *S. coelicolor* promoter region of *nrdAB* (class I RNR) or *nrdRJ* (NrdR followed by class II RNR) (Fig. 1b, c). Unexpectedly, a combination of dATP and ATP resulted in strong binding to the *nrdRJ* promoter and somewhat weaker binding to the *nrdAB* promoter, suggesting a specific and differentiated binding (Fig. 1b). A combination of dATP and ADP also resulted in binding to the *nrdRJ* and *nrdAB* promoter regions (Fig. 1b). Other adenosine nucleotides or combinations thereof showed very low affinity of NrdR to DNA (Supplementary Figs. 2, 3). Our findings clearly challenge earlier proposed mechanisms where it was suggested that dATP-loaded NrdR was sufficient for DNA binding[10,11,16], or that the monophosphate forms of ATP or dATP, i.e., AMP or dAMP, promoted DNA binding[12]. Our studies clearly demonstrate that DNA binding requires specific combinations of nucleotides.

**ATP-loaded NrdR forms a dodecamer**. The structure of ATP-loaded NrdR (2.96 Å resolution from two merged cryo-EM datasets; Supplementary Table 1) reveals a dodecameric assembly of three tetramers (Fig. 2a, Supplementary Figs. 4, 5, and Supplementary Movie 1). 2D classification and particle orientation distribution for this dataset are shown in Supplementary Fig. 6, revealing a homogeneous population of NrdR dodecamers. In each tetramer, the ATP-cones of chains A and B, and those of chains C and D, interact with each other, respectively. For the Zn-ribbon, the domains of chains A and D and those of chain B and C interact, which generates an intertwined tetrameric structure (Fig. 2b). In addition, the orientation between the ATP-cone and the Zn-ribbon domains differs by ~90° in chains A and C compared to chains B and D (Fig. 2b). Perhaps related to these differences, the ATP occupancy in chains B and D is close to 100%,

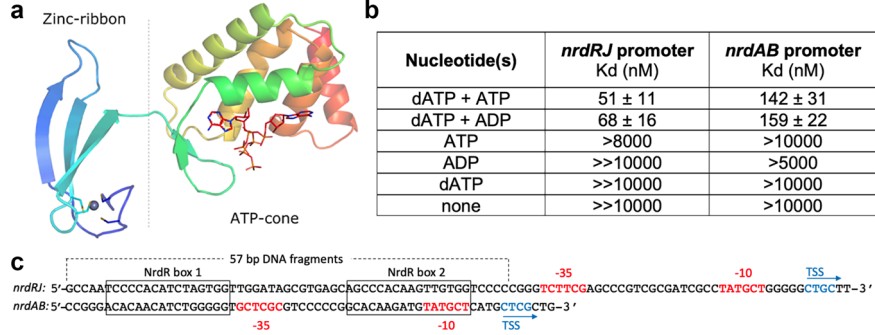

**Fig. 1 Structure and binding affinity of NrdR. a** Structure of the ATP-loaded NrdR monomer (chain A) colored as a rainbow from dark blue at the N-terminus to red at the C-terminus. The probable position of the Zn is indicated with a gray sphere. **b** Binding constants of NrdR to *nrdRJ* and *nrdAB* promoter regions determined by microscale thermophoresis (MST) as shown in Supplementary Figs. 2 and 3. $K_D$s between 10 and 19 µM are denoted >10 µM, and above 19 µM as ≫10 µM. **c** Promoter regions of *nrdRJ* and *nrdAB* with NrdR boxes, RNA polymerase binding sites (red), transcription start sites (blue; from Borovok et al. 2004 (ref. [13])), and 57 bp DNA fragments used in MST analyses indicated.

| Nucleotide(s) | *nrdRJ* promoter Kd (nM) | *nrdAB* promoter Kd (nM) |
|---|---|---|
| dATP + ATP | 51 ± 11 | 142 ± 31 |
| dATP + ADP | 68 ± 16 | 159 ± 22 |
| ATP | >8000 | >10000 |
| ADP | ≫10000 | >5000 |
| dATP | ≫10000 | >10000 |
| none | ≫10000 | >10000 |

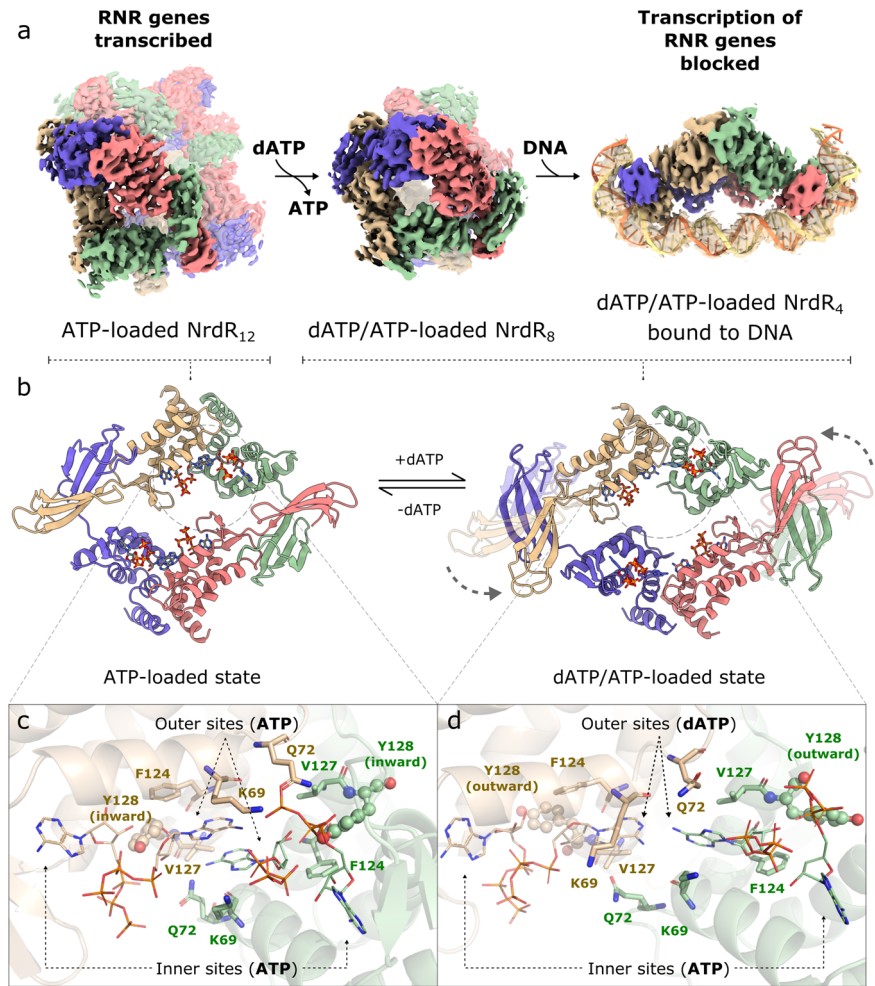

**Fig. 2 Mechanism of NrdR function involving dodecameric, octameric, and tetrameric structures. a** Surface representation of the cryo-EM maps for the dodecameric, octameric, and DNA-bound tetrameric NrdR structures. **b** Cartoon representation of the ATP-loaded NrdR tetramer (left) and the dATP/ATP-loaded tetramer (right). Chains A, B, C, and D are colored in beige, green, pink, and blue, respectively. **c** Interface between the ATP-cones in chain A (beige) and chain B (green) in the ATP-loaded dodecamer and (**d**) in the dATP/ATP-loaded tetramer. Panels **c**, **d** were made from the same perspective, based on an alignment of the ATP-cones in chains A and B in both structures.

but lower in chains A and C (Supplementary Fig. 7c). The described conformational differences allow the tetramers to assemble into a dodecamer, with occluded Zn-ribbon domains, unable to bind DNA. Supplementary Fig. 8 illustrates the different Zn-ribbon orientations between NrdR chains within the ATP-loaded structure, as well as in the dATP/ATP-loaded structures. Analysis of interactions between the protein chains using PISA[27] shows that contacts between pairs of ATP-cones and pairs of Zn-ribbon domains bury ~8100 Å$^2$ of solvent-accessible area (SAA), while each tetramer buries ~4900 Å$^2$ of SAA through contacts with the two other tetramers. Based on a positive predicted free energy of dissociation of ~60 kcal/mol, PISA classifies the dodecamer as stable in solution.

**Structure of tetrameric NrdR bound to DNA**. The structure of dATP/ATP-loaded NrdR in complex with the *nrdRJ* promoter (3.3 Å resolution; Supplementary Table 1) reveals an NrdR tetramer bound to the dsDNA (Fig. 2a and Supplementary Movie 2). 2D classes and particle orientation distribution for this dataset are shown in Supplementary Fig. 6a, showing a homogeneous population of dsDNA-bound NrdR tetramers. The relative orientations of the dimeric Zn-ribbon domains and the ATP-cone domains differ between this tetramer and the ATP-loaded tetramers that

build up the dodecamer (Fig. 2b and Supplementary Fig. 8). The binding of NrdR to DNA (Fig. 3a) is mediated by a pair of Zn-ribbon domains interacting with the phosphate backbones at each NrdR box via six arginines from each domain, including the highly conserved RRRR motifs (Supplementary Fig. 1). This results in a close to 90° kink[28] in the DNA at each NrdR box at the weakly stacking AT bases (Fig. 3). The minor groove phosphate backbones in the kink regions are only about 5 Å apart at their closest approach. The short distance between the DNA-binding regions of the two Zn-ribbons, in each Zn-ribbon pair, is the main cause of the compression of the minor groove. The DNA kink leads to strongly distorted base-pairing compared to canonical B-form DNA and to several exposed bases that interact with NrdR. Two amino acids that appear to provide DNA sequence specificity are D15 and R17 on each Zn-ribbon, which form an intricate network with the exposed bases in the palindromic NrdR boxes, where the GC base pairs at positions 9 and 18 in box 1, and 40 and 49 in box 2 are highly conserved (Figs. 3c, 4). As the dATP/ATP-loaded NrdR tetramer contains two DNA-binding sites 80 Å apart, it is suited to bind to two NrdR boxes located approximately three dsDNA helix turns from each other, also contributing to the specificity. NrdR binding facilitates intramolecular interactions within the DNA that are not possible in canonical B-DNA, further stabilizing the DNA-NrdR complex (Fig. 3).

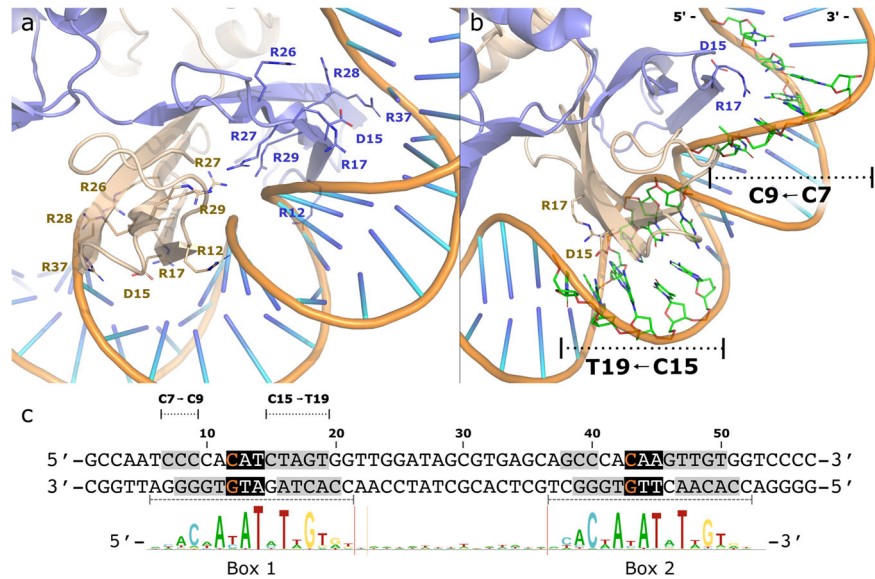

**Fig. 3 Interactions between NrdR and DNA. a** Chains A (beige) and D (blue) of the DNA-bound dATP/ATP-loaded NrdR bound to NrdR box 1. Arginine residues 12, 26–29, and 37 interact with the DNA phosphate backbone. **b** D15 and R17 are located close to the DNA bases in the major groove. **c** DNA sequence of the *S. coelicolor nrdRJ* promoter region (top) and sequence logo of identified NrdR boxes (bottom). Nucleotides interacting with NrdR (shaded gray); highly distorted interactions, with cross-base pair interactions (orange on black); middle of sharp bend (white on black).

| *nrdRJ* promoter (base pair changes) | Type of NrdR protein | | | |
|---|---|---|---|---|
| | wild type $K_D$ (nM) | D15A $K_D$ (nM) | R17A | D15A/R17A |
| wild type | 53 ±9 | 50 ±10 | no binding | no binding |
| G9/C18 (box 1 inverted) | 2 600 ±560 | 224 ±27 | no binding | - |
| C18/C49 (box 1 and box 2 single inversions) | 3 100 ±730 | 314 ±44 | no binding | - |
| T9/T18 (box 1 mutated) | 5 300 ±1 300 | 54 ±14 | no binding | - |
| T40/T49 (box 2 mutated) | 8 600 ±1 000 | 110 ±21 | no binding | - |
| T9/T18/T40/T49 (box 1 and box 2 mutated) | »>10 000 | 523 ±108 | no binding | - |

```
            9        18                              40        49
5’−GCCAATCCCCACATCTAGTGGTTGGATAGCGTGAGCAGCCCACAAGTTGTGGTCCCC−3’
3’−CGGTTAGGGGTGTAGATCACCAACCTATCGCACTCGTCGGGGTGTTCAACACCAGGGG−5’
            |____Box 1____|                     |____Box 2____|
```

**Fig. 4 Importance of highly conserved GC base pairs in the *nrdRJ* NrdR boxes and two conserved residues in the NrdR protein.** Upper panel: Results of MST experiments performed in the presence of 1 mM dATP and 1 mM ATP. A binding constant of »10 µM indicates a $K_D$ between 19 and 96 µM; a minus sign means that no experiment was performed. Lower panel: Mutated base pairs (red) in the 57 bp *nrdRJ* promoter.

At this resolution, we cannot confidently assign the identity of the individual bases in the DNA, when looking at them individually. However, we can distinguish the general structure of almost the entire DNA construct (50 of 57 bases). This, together with the twofold symmetry in the DNA and in the NrdR tetramer, puts restraints on the registry of the DNA, in relation to the NrdR tetramer. These structural considerations, in combination with mutational analysis of both the protein and the DNA construct, support the register of the DNA. We are confident in the side-chain assignments in the interface between DNA and NrdR. However, the precise orientation of the side chains of some amino acids is somewhat ambiguous because of the limited local resolution. The structure of DNA-bound NrdR suggests that this tetrameric repressor prevents transcription of RNR both by directly blocking access to the promoter region as well as restricting access for RNA polymerase in the sharp 180° turn of the DNA (Fig. 2a).

**Mutations supporting DNA–protein interactions.** We performed two sets of experiments to support the specific DNA–protein interactions suggested in the cryo-EM structure of tetrameric NrdR bound to the *nrdRJ* promoter region. One set involved mutations in the highly conserved GC base pairs in the NrdR boxes in the *nrdRJ* promoter DNA and the other set involved mutations in the conserved residues D15 and R17 in NrdR. Inversion of the GC base pairs at positions 9 and 18 results in a 50-fold decreased binding of NrdR, and the inversion of one GC base pair in each NrdR box results in close to 60-fold decreased binding (Fig. 4 and Supplementary Fig. 9a). More drastic changes of the highly conserved GC base pairs result in more than 100-fold lower binding affinities. The D15A NrdR mutant binds strongly to the wild-type *nrdRJ* promoter DNA, and surprisingly also strongly to all mutant DNAs, with 10- to 100-fold stronger binding compared to wild type (Fig. 4 and Supplementary Fig. 9). The D15A NrdR also binds with a $K_D$ of 2600 ± 300 nM to the negative control promoter *cydA* (Supplementary Fig. 9b), to which both wild-type NrdR and the R17A mutant do not bind at all (Supplementary Figs. 2, 10a). The R17A NrdR mutant does not bind to the wild-type or mutant *nrdRJ* promoter DNA, and this is also the case for the D15A/R17A double mutant (Fig. 4 and Supplementary Fig. 10). Our results

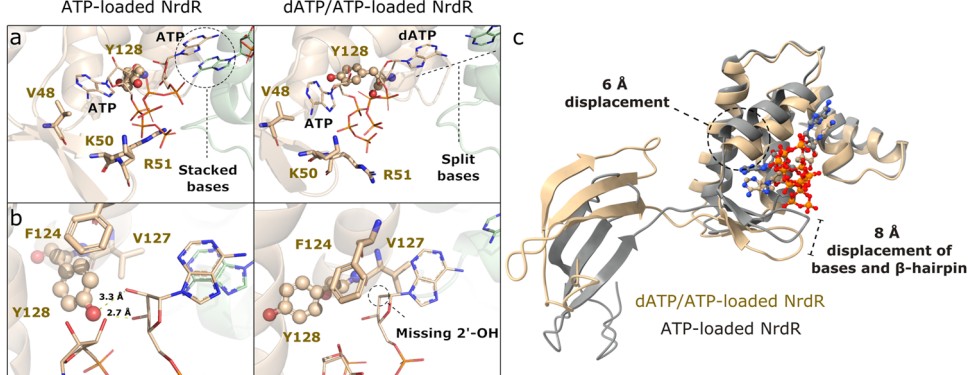

**Fig. 5 Nucleotide binding to NrdR. a** Nucleotide-binding site of the ATP-loaded NrdR dodecamer (left) and dATP/ATP-loaded NrdR tetramer (right). Chain A is colored in beige and chain B is colored in green. **b** Close-up illustration of the displacement of Y128 (highlighted in ball-and-stick view) between ATP-loaded NrdR (left) and dATP/ATP-loaded NrdR (right). **c** ATP-loaded NrdR chain A (gray) aligned with dATP/ATP-loaded NrdR chain A (beige), highlighting the most prominent displacements between them. The alignment was made with the ATP-cone domains only.

clearly support the assumptions from the cryo-EM structure of DNA-bound NrdR that conserved residues D15 and R17 are important for the specific binding of NrdR to the *nrdRJ* promoter sequence. In addition, an earlier study in *E. coli* indicated that mutation of the conserved RRRR motif in NrdR resulted in impaired DNA binding[29].

**dATP/ATP-loaded NrdR forms an octamer**. The structure of dATP/ATP-loaded NrdR in the absence of DNA (3.1 Å resolution) reveals an octameric complex composed of two tetrameric assemblies (Fig. 2a and Supplementary Movie 3). 2D classification of this dataset showed a particle population consisting predominantly of NrdR octamers, with a small population of NrdR tetramers (Supplementary Fig. 6b). Apart from a slight adjustment of the angles of the Zn-ribbon domains (Supplementary Fig. 8) the two tetramers in the octamer are virtually identical to the tetramer that interacts with DNA (Fig. 2b). Tetramers competent to bind DNA are locked in an octameric form in which the Zn-ribbons of two tetramers face each other. The octamer would therefore need to dissociate to bind DNA. PISA analysis reveals that ~3700 Å² of SAA are buried in four distinct small interaction areas between tetramers, compared to ~8550 Å² buried within each tetramer. This implies that interactions between tetramers are significantly weaker than those within tetramers. The predicted free energy of dissociation of about −13 kcal/mol suggests that the octamer is unstable or marginally stable in solution, thus liable to dissociate into tetramers. This oligomerization mechanism may provide an additional layer of regulation by controlling the concentration of tetrameric NrdR, available for binding the promoters.

**Nucleotide coordination differences between assemblies**. In the NrdR ATP-cone, nucleotides bind to two sites, an "inner" site similar to that seen in all ATP-cones to date and an "outer" site found only in ATP-cones that bind two nucleotides. The two nucleotides bind with their triphosphate tails oriented towards each other, as previously seen in crystal structures of those ATP-cones[22,24]. This binding mode requires an $Mg^{2+}$ ion for charge neutralization to allow the tails to come into proximity. No such $Mg^{2+}$ ion is visible in the cryo-EM maps, perhaps due to limited resolution, but we assume that it is present, as $Mg^{2+}$ is strictly necessary for nucleotide binding to NrdR. In both the ATP- and dATP/ATP-loaded structures, K50, R51, and E56 contact the nucleotide in the inner binding site, which always contains ATP, and K62 contacts the nucleotide in the outer site, which can bind

either ATP or dATP (Supplementary Fig. 4). In earlier studies, mutation of V48, K50, E56, or K62 impaired nucleotide binding, changed the oligomeric state of NrdR, and abolished DNA binding[11]. Likewise, mutation of K53 in *E. coli* NrdR, the equivalent of *S. coelicolor* K50, also abolished DNA binding[12]. The inner site is highly specific for ATP. Sterically the pocket is dimensioned for a purine base, and adenine is specifically recognized, as in previously-characterized ATP-cones, by two hydrogen bonds from the main chain of the loop between the β-hairpin and the first helix of the ATP-cone that match the 6-$NH_2$ group and the unprotonated ring N1 atom in adenosine. There is also an H-bond from E56 to the 6-$NH_2$ group. Replacing adenosine by guanosine would result in three H-bonding mismatches and also a steric clash between the 2-$NH_2$ group and F58 in the same loop. For the outer site, some features also appear very specific for ATP/dATP. In the octamer, the 6-$NH_2$ group of dATP makes an H-bond to the main-chain carbonyl group of a neighboring monomer. Why the outer site of the dodecamer is specific for adenosine nucleotides is less clear, as the two adenosine rings of ATP stack on each other (Fig. 5a).

How can the small difference between ATP and dATP of a single 2′ hydroxyl group translate into the massive structural rearrangements of NrdR from a dodecamer into an octamer that ultimately dissociates to a DNA-binding tetramer? The base and ribose of the inner site ATP make essentially the same interactions in both forms and the conformational changes appear driven largely by the 2′-OH group in the outer site. The absence of the 2′-OH in dATP allows a close interaction of the ribose with F124, V127, and the backbone of Y128 (Figs. 2d, 5b and Supplementary Fig. 4). This is not possible in the ATP-bound NrdR due to the 2′-OH group (Fig. 2c and Supplementary Fig. 4) and the outer site ATP shifts further out of the site. This allows a dramatic 180° flip of Y128, which faces away from the nucleotide-binding site in the dATP/ATP-loaded structures but H-bonds to the 2′-OH and 3′-OH of the outer ATP in the ATP-loaded structure (Fig. 5b). These differences are coupled with a hinge movement of up to 6 Å in the last two α-helices of the ATP-cones (Supplementary Movies 4, 5). As the movement happens in both monomers of the dimer, this results in a concerted rearrangement of the interface between the two ATP-cones in each dimer (Fig. 2b, c). As a result, the outer nucleotide of the second monomer moves by around 8 Å towards the first monomer (Fig. 5a and Supplementary Movies 4, 5). The bases of the outer site nucleotides stack on each other in the ATP-loaded form rather than being sandwiched between the side chains of K69 and V127 and making end-to-end contacts in the dATP/ATP-loaded

form (Fig. 5a, c and Supplementary Fig. 4). In its "flipped-in" position in the ATP-loaded form, the hydroxyl group of Y128 also coordinates the inner-site ATP (Fig. 2c). Interestingly, a Y128A mutation resulted in NrdR co-purifying with two dATP molecules per mole protein[11]. The rearrangement of the dimer interface also results in the coordination of the phosphate groups of both nucleotides by K69 from the adjacent ATP-cone of the dodecamer, which is too distant in the dATP/ATP-loaded octamer (Fig. 2c). Q72 from the adjacent monomer shifts from coordinating the α-phosphates of the outer nucleotide to that of the inner nucleotide.

The flip of Y128 and the stacking of the adenosine bases in the ATP-bound form leads to a distinct displacement of the β-hairpin that carries K50 and R51. These residues (K50 and R51) directly bind the phosphate groups of the nucleotide and are located on one side of the β-hairpin (Fig. 5a, c). The other side of this β-hairpin constitutes one of the key interactions in the multi-merization of the dodecamer. The move of the hairpin changes the oligomerization interfaces of NrdR. However, there are many changes that directly and indirectly depend on the absence or presence of the 2′-hydroxyl group, and it is, therefore, difficult to separate cause and effect in the intricate structural rearrangements caused by ATP versus dATP.

**The mechanism of NrdR oligomerization.** In the absence of nucleotides, *S. coelicolor* NrdR elutes mainly as a dimer (44 kDa) in gas-phase electrophoretic mobility molecular analysis (GEMMA; Fig. 6). In contrast, the addition of dATP, dADP, ADP, or a combination of dATP and ADP results in a tetramer (90 kDa). The major fraction in the presence of ATP is a dodecamer (247 kDa), and a combination of dATP and ATP results in an equilibrium between an octamer (160 kDa) and a tetramer. In the presence of dAMP and cyclic di-AMP, NrdR is in a dimer-tetramer equilibrium. We verified the GEMMA results using analytical size-exclusion chromatography (Supplementary Fig. 11) using close to physiological nucleotide concentrations of dATP and ATP. A mutant NrdR lacking the Zn-ribbon shows that the isolated ATP-cone can only form dimers, but not higher oligomers in presence of nucleotides (Supplementary Fig. 11), suggesting that the tetrameric unit that forms both in the ATP- and dATP/ATP-loaded oligomers is mediated by the pairwise Zn-ribbon interactions. This is confirmed by PISA analysis: in all oligomers, the Zn-ribbon dimers bury ~1200 Å² of the solvent-accessible area while the ATP-cone dimers bury only ~900 Å².

**A multifactorial sensor for nucleotides defines a new ATP-cone class.** It was earlier suggested that hydrolysis of ATP and/or dATP to monophosphates enables NrdR to bind to DNA[12]. We,

therefore, measured the binding of nucleotides to *S. coelicolor* NrdR using isothermal calorimetry (ITC) (Supplementary Fig. 12). The strongest binding was observed with ADP and dADP ($K_{DS}$ of 0.6 and 0.4 µM, respectively). ATP is more weakly bound compared to dATP ($K_{DS}$ of 4 and 1 µM, respectively) and AMP, dAMP, and di-cyclic AMP were comparable to ATP in binding strengths. NrdR could not bind any other dNTP than dATP. Binding curves for all tested nucleotides were consistent with a single set of binding sites. We interpret these results to show the high-affinity binding of ATP to the inner site of the NrdR ATP-cone and dATP to the outer site for the following reasons. Half of the monomers in the ATP-bound dodecamer have lower occupancy of ATP to the outer site compared to the inner site (Supplementary Fig. 7c), and all monomers in the dATP/ATP-loaded octameric and tetrameric structures have full occupancy in both sites (Supplementary Fig. 7c). Given an intracellular ATP concentration in the mM range[30], the inner site of the NrdR ATP-cone would be occupied by ATP. Despite having lower affinity, the outer site would plausibly also be occupied by ATP but will be replaced by dATP when its cellular concentration increases, e.g. as a result of low DNA synthesis. The dATP/ATP-loaded NrdR will then bind to RNR promoters and restrict transcription of the RNR genes. We assume that when the intracellular concentration of ADP increases at the expense of ATP, for example under conditions of phosphate limitation[30] and in sporulated cells[31], ADP with its 4–5 times stronger $K_D$ will replace ATP at the inner site, and the dATP/ADP-loaded NrdR will restrict RNR transcription at low cellular energy levels.

The *S. coelicolor* NrdR ATP-cone defines a new class of ATP-cones that binds two different nucleotides simultaneously. In this respect, it most closely resembles the RNR ATP-cones with two nucleotide-binding sites previously observed in the structures of *P. aeruginosa* NrdA[22] and *Leeuwenhoekiella blandensis* NrdB[24] more than the type in eukaryotic RNR that binds only a single nucleotide[19] (Supplementary Fig. 13a). Recently it was suggested that the ATP-cone in *Aquifex aeolicus* RNR binds two ATP and that the well-characterized *E. coli* RNR ATP-cone, thought to bind only one nucleotide, has a "cryptic" second binding site for ATP[32,33]. In the previously-described ATP-cone, one nucleotide binds to the inner site (Supplementary Fig. 13b) with its adenosine base contacting a β-hairpin defining the "roof" of the ATP-cone and the following loop, as also observed in NrdR. The outer site differs from the one in NrdR in that the base of the nucleotide stacks with an aromatic residue in the C-terminal helix of the ATP-cone. Its triphosphate moiety makes a "tail-to-tail" interaction with the inner-site nucleotide, though differently from what is seen in NrdR. This type of ATP-cone appears to specialize in self-interactions through its C-terminal helices[22]. The C-terminal region in NrdR is longer, forming a helix-loop-helix motif, and the second helix folds back against the first to generate a 4-helix bundle when the ATP-cones dimerize (Supplementary Fig. 13c). These interactions drive a relative orientation of the two ATP-cones that project the bases of the outer site nucleotides so closely together that they form direct H-bonds (octamer or tetramer) or stack with each other (dodecamer).

**Mechanism of action of NrdR.** Our combined structural and biochemical studies reveal a mechanism of action of NrdR that differs from those suggested earlier[10,11,16] where one molecule of ATP or dATP, was believed to bind per NrdR monomer, and an octameric or a dimeric NrdR was believed to bind to DNA. A significantly more complex mechanism was also suggested in which both ATP and dATP inhibited DNA binding[12]. In this case binding of NrdR to DNA was only enabled after hydrolysis of the nucleotides to monophosphates by unknown mechanisms. In

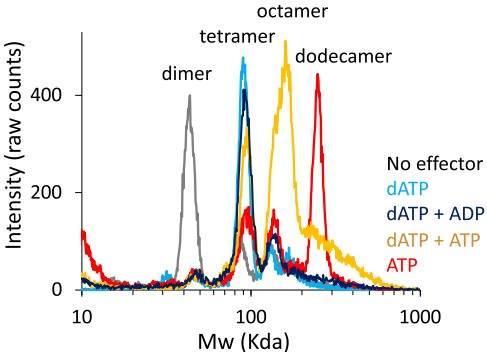

**Fig. 6 Oligomeric states of NrdR.** Oligomeric states of *S. coelicolor* NrdR loaded with nucleotides determined by GEMMA. Gray: no effector, red: ATP, light blue: dATP, orange: dATP + ATP, dark blue: dATP + ADP.

contrast to all earlier suggested mechanisms, we show that the NrdR monomer contains two nucleotide-binding sites, one inner site that binds ATP and one more exposed site that can bind either ATP or dATP. As illustrated in Fig. 2a, ATP-loaded NrdR forms a dodecamer that cannot bind to DNA. When dATP levels increase as a result of decreased DNA synthesis, dATP competes out ATP from the outer site in the ATP-cone and a dATP/ATP-loaded octamer forms. The octamer is in equilibrium with a tetramer that has the capacity to bind to the NrdR boxes in the RNR promoter regions, thereby inhibiting the expression of RNR genes. The complex between NrdR and DNA combined with studies of mutant NrdR proteins and mutant promoter regions reveal the key determinants for the sequence specificity. Interestingly, ADP binds to the inner site almost five times stronger compared to ATP, and dATP/ADP-loaded NrdR binds to DNA with equal affinity compared to dATP/ATP-loaded NrdR, indicating that a low cellular energy state, as well as low consumption of DNA building blocks, triggers the repressor activity of NrdR. Finally, to our knowledge, this is the only example of a mobile protein domain, the ATP-cone, that both controls the expression of an enzyme and allosterically regulates the activity of the same enzyme.

## Methods

**Bioinformatics**. To identify NrdRs in organisms, we searched an in-house database consisting of Prokka[34] annotated genome assemblies selected for the 05-RS95 release of the Genome Taxonomy Database (GTDB)[35] as well as NCBI's RefSeq database (downloaded 20201015)[36] with an HMMER profile[37] representing known NrdR sequences. The GTDB results were used to form a picture of NrdR presence in bacteria and archaea, using only the species-representative genomes to get as fair a picture of the distribution as possible. The NCBI data was used to identify NrdR presence in eukaryotes, without filtering any that could be confirmed as not caused by bacterial contamination. The identified sequences were clustered with USEARCH[38] at an identity threshold of 0.5 to reduce redundancy and manually inspected to remove fragmental and low-quality sequences ($n = 326$). Subsequently, the sequences were aligned with Probcons (v.1.12)[39] and a logo was created using the Skylign online service[40].

To identify NrdR boxes in organisms, we used known NrdR box sequences to build an HMMER profile[37] covering the two NrdR boxes and the linker sequence between them. The HMM profile was used to search GTDB (release 05-RS95)[35] species-representative genomes and the results were used iteratively to enhance the HMM profile and search the genomes again, in order to cover more bacterial phyla. We used Probcons (v.1.12)[39] to align the sequences, and low-quality sequences were manually eliminated. The final HMM profile included 935 sequences. The logo was built using the Skylign online service[40].

**Plasmids**. The *nrdR* gene from *S. coelicolor* strain M145 and a truncated mutant lacking both N- and C-terminal domains (41 and 32 amino acids respectively) were amplified and cloned into pET30a(+) vector (Novagen) as described previously[7] resulting in pET30a(+)::*nrdR* and pET30a(+)::*nrdRΔNΔC* constructs expressing wild-type and mutant NrdR proteins with C-terminal hexahistidine (His) tags. To obtain NrdR bearing point mutations D15A, R17A and the double mutant D15A/R17A, constructs pET30a(+)::ScoNrdR_D15A, pET30a(+)::ScoNrdR_R17A and pET30a(+)::ScoNrdR_D15A/R17A containing nucleotide mismatches A44C/C45A, C49G/G50C/T51G and A44C/C45A/C49G/G50C/T51G respectively and C-terminal hexahistidine (His) tags, were ordered from GenScript.

**Reagents**. ATP, dATP, dTTP, dGTP, and dCTP were purchased from Thermo Fisher Scientific as 100 mM solutions, pH 7. Other nucleotides were purchased from Sigma-Aldrich. Stock solutions of 50–250 mM were prepared in water and the pH was adjusted to 7.

**Protein expression**. Overnight cultures of *E. coli* BL21(DE3) bearing pET30a(+)::*nrdR*, pET30a(+)::*nrdRΔNΔC*, pET30a(+)::ScoNrdR_D15A, pET30a(+)::ScoNrdR_R17A, and pET30a(+)::ScoNrdR_D15A/R17A were diluted to an absorbance at 600 nm of 0.1 in LB (Luria-Bertani) liquid medium, containing kanamycin (50 μg/ml) and shaken vigorously at 37 °C. At an absorbance of 0.8 at 600 nm, isopropyl-β-D-thiogalactopyranoside (Sigma) was added to a final concentration of 0.4 mM; Zn(CH₃CO₂)₂ was added to 0.1 mM to all cultures except culture bearing pET30a(+)::*nrdRΔNΔC*. The cells were grown overnight under vigorous shaking at 20 °C and harvested by centrifugation. The cell pellet was stored at −80 °C.

**Protein purification**. All buffers used for purification of wild-type and mutant NrdR proteins contained 2 mM dithiothreitol (DTT) or 0.1 mM Tris(2-carboxyethyl)phosphine (TCEP). The cell pellet was thawed and resuspended in lysis buffer: 50 mM Tris-HCl pH 7.6 containing 300 mM NaCl, 10% glycerol, 10 mM imidazole, and 2 mM DTT or 0.1 mM TCEP. Phenylmethylsulfonyl fluoride (PMSF) was added to 1 mM to the cell suspension, the mixture was sonicated in an ultrasonic processor (Misonics) until clear and the lysate was centrifuged at 18,000 × *g* for 45 min at 4 °C. The recombinant His-tagged protein was first isolated by metal-chelating affinity chromatography using an ÄKTA prime system (Cytiva): the supernatant was loaded on a HisTrap FF Ni-Sepharose column (Cytiva) equilibrated with lysis buffer (w/o PMSF), washed thoroughly with buffers containing 10 and 60 mM imidazole and eluted with buffer containing 500 mM imidazole. The proteins were desalted using HiPrep 26/10 Desalting column (Cytiva) in a buffer containing 50 mM Tris-HCl pH 7.6, 300 mM NaCl, 10% glycerol, and 2 mM DTT or 0.1 mM TCEP, frozen and stored at −80 °C. NrdRΔNΔC was further purified by loading on fast protein liquid chromatography (FPLC) using HiLoad 16/600 Superdex 200 pg column equilibrated with a buffer containing 50 mM Tris-HCl pH 8, 300 mM NaCl, and 5 mM DTT. To remove bound nucleotides the proteins were subjected to hydrophobic interaction chromatography using the HiTrap Phenyl FF (high sub) column (Cytiva) in 50 mM Tris-HCl, pH 7.6 or pH 8.5, 1 and 0.75 M (NH₄)₂SO₄ for NrdR and NrdRΔNΔC respectively, and 2 mM DTT or 0.1 mM TCEP, washed extensively (70–100 column volumes) with the same buffer, and eluted with buffer without ammonium sulfate followed by elution with water. Protein recovery after this stage was ~20%. Buffer exchange was performed using the HiPrep 26/10 Desalting column, alternatively PD10 column (Cytiva), or by addition of buffer components to the sample eluted with water. Buffer normally contained 50 mM Tris-HCl, pH 7.6 (or 8.5 for NrdRΔNΔC), 300 mM NaCl, 10% glycerol, 2 mM DTT. Buffer for the NrdR sample used for ITC contained 50 mM Tris-HCl pH 8, 300 mM NaCl, 10% glycerol, and 1 mM TCEP. Protein concentration was determined using Coomassie Plus (Bradford) Assay Kit (Thermo Fisher Scientific) using a BSA standard curve. The apo-NrdR was aliquoted, frozen in liquid nitrogen, and stored at −80 °C until used at concentrations of ~1–2 mg/ml. The apo-NrdR cannot be further concentrated without the addition of effector nucleotides due to apo-protein instability at higher concentrations. Consistent with a previously reported value[7] the zinc content of NrdR was 0.78 ± 0.04 mol Zn/mol protein (based on three independently prepared samples) quantified using total-reflection X-ray fluorescence (TXRF) on a Bruker PicoFox S2 instrument and analyzed with software provided with the spectrometer. A gallium internal standard at 2 mg/l was added to the samples (v/v 1:1) before the measurements. To obtain high protein concentrations of the dATP/ATP-loaded forms of wild-type and mutant D15A, R17A, and D15A/R17A NrdR the purified proteins after metal-chelating affinity chromatography and desalting were supplemented with 10 mM MgCl₂, 1 mM ATP, 1 mM dATP, and 2 mM DTT and concentrated using Vivaspin 500 ultrafiltration units (50,000 MWCO PES membrane, Sartorius).

**Isothermal titration calorimetry**. ITC experiments were carried out on a MicroCal ITC 200 system (Malvern Panalytical) in a buffer containing 50 mM Tris-HCl, pH 8, 300 mM NaCl, 10% glycerol, 1 mM TCEP, and 10 mM MgCl₂. Measurements were done at 20 and 10 °C with a stirring speed of 1000 rpm. The initial injection volume was 0.6–1 μl over a duration of 1.2–2 s. All subsequent injection volumes were 2–2.5 μl over 4–5 s with a spacing of 150–180 s between the injections. Data for the initial injection were not considered. The concentration of NrdR in the cell was 10 or 20 μM. In the syringe, ATP, dATP, cyclic di-AMP, dTTP, dGTP, and dCTP concentrations were in the range of 120–200 μM, ADP and dADP 65–150 μM, and AMP and dAMP 200–420 μM. For titration of cAMP, the NrdR concentration was 60–96 μM and cAMP 0.7–1 mM. The data were analyzed using the one set of sites model of the MicroCal PEAQ-ITC analysis software (Malvern Panalytical). Standard deviations in thermodynamic parameters, N and $K_D$ were estimated from the fits of three different titrations. Attempts to fit the data to a two-sites model gave too high uncertainty.

**Microscale thermophoresis**. We studied the binding of *S. coelicolor* NrdR to its cognate *nrdAB* (class I) and *nrdRJ* (class II) recognition sites in the RNR promoter regions using microscale thermophoresis (MST)[41]. Oligonucleotides of 57 base pairs containing the NrdR boxes and flanking regions of five nucleotides on each side, and a negative control *S. coelicolor cydA* gene promoter region were ordered from Genscript and from Sigma-Aldrich (Supplementary Table 2). All oligonucleotides were purified by HPLC by the manufacturer and the sense oligonucleotides were fluorescently labeled by the Cy5 dye at their 5′ end by the manufacturer. For *nrdAB* another pair of oligonucleotides was ordered, in which the antisense oligonucleotide was labeled with Cy5 at its 5′ end (Supplementary Table 2), resulting in the Cy5 label being proximal to NrdR box 2 instead of box 1 as in the rest of the oligonucleotide pairs. Mutated *nrdRJ* oligonucleotides, containing Cy5 dye at the 5′ end of the sense oligonucleotide are shown in Supplementary Table 2. Freeze-dried oligonucleotides were reconstituted in 50 mM Tris pH 8, 50 mM NaCl and 1 mM EDTA. 50 and 57.5 pmol of labeled and unlabeled oligonucleotides respectively were mixed in a total volume of 50 μl containing 50 mM Tris-HCl pH 7.4, 50 mM NaCl and annealed using a thermoblock. The annealing program

included incubation for 5 min at 95 °C and gradual cooling to 25 °C using 140 cycles of −0.5 °C and 45 s per cycle, resulting in 1 μM double-stranded DNA.

MST was performed using the Monolith NT.115 instrument (NanoTemper Technologies GmbH). Binding was measured between NrdR and double-stranded oligonucleotides containing the promoter region of *nrdAB* or *nrdRJ* labeled with Cy5 in MST buffer containing 25 mM Tris-HCl pH 7.6, 150 mM NaCl, 10 mM MgCl$_2$, 5% glycerol, 1 mM DTT, 0.05% Tween-20, and the specified nucleotide or a combination of nucleotides (1 mM each, unless otherwise stated) at room temperature. A 16-step dilution series was prepared by adding 10 μl buffer to 15 tubes. Wild-type NrdR, 20 μl of 10–117 μM (32 μM in most of the experiments), was placed in the first tube, and 10 μl was transferred to the second tube and mixed well by pipetting (1:1 dilution series). To each tube of the dilution series 10 μl of the binding partner (double-stranded Cy5-labeled oligonucleotide in the same buffer) was added to reach a final concentration of 10 nM. The samples were incubated for 5 min at room temperature before being transferred to Monolith™ NT.115 Series Capillaries (NanoTemper Technologies GmbH). The capillaries were scanned using the MST instrument (40–60% excitation power, medium MST power). Negative controls included the binding of NrdR to a double-stranded oligonucleotide containing the promoter region of *cydA* gene as well as the binding of bovine serum albumin (BSA to the promoter regions of *nrdAB* in the presence of selected nucleotides. The binding of NrdR to a single-stranded Cy5-labeled *nrdAB* oligonucleotide was also tested with a negative result. Obtained MST data was analyzed and fitted using the MO. Affinity Analysis v2.3 software (NanoTemper Technologies) with default parameters. For NrdR loaded with dATP + ATP and dATP + ADP nucleotide combinations K$_D$ and standard deviation were calculated using fits from three individual titrations. The K$_D$s for NrdR loaded with single nucleotides and other nucleotide combinations could not be reliably determined since the titration curves did not reach a plateau even when high NrdR concentrations (117 μM) were used and therefore are only estimates. To test the binding of wild-type NrdR to mutated *nrdRJ* oligonucleotides in the presence of 1 mM ATP and 1 mM dATP, NrdR at a concentration of 32 or 684 μM was used. Using high protein concentrations did not affect the binding of wild-type NrdR to wild-type oligonucleotides but was necessary for a reliable determination of K$_D$ using the mutant oligonucleotides. NrdR mutant D15A was used at concentrations of 108, 114, and 173 μM (binding to *nrdRJ* promoter) and 114–927 μM (binding to cydA promoter); R17A and D15A/R17A mutants were used at concentrations of 114–660 μM and 87–472 μM, respectively.

**Cryo-electron microscopy**. A cryo-EM processing summary is presented in Supplementary Fig. 5. To prepare NrdR samples for cryo-EM containing ATP-loaded NrdR, the apo-protein at 1.4 mg/ml was thawed, 10 mM MgCl$_2$, and 1 mM ATP were added, and the sample was incubated for 10 min at 7 °C. The sample was loaded on an FPLC system using a Superdex 200 Increase 10/300 GL column equilibrated with a buffer containing 50 mM Tris-HCl pH 8.0, 150 mM NaCl, 10 mM MgCl$_2$, 1 mM TCEP, and 0.5 mM ATP. The protein was eluted as a single peak and 0.25 ml fractions were collected. Fractions from the middle of the peak at a concentration of 0.4 mg/ml (corresponding to 19 μM monomeric NrdR) were used to freeze cryo-EM grids, which were prepared by applying 3 μl of sample on Quantifoil R1.2/1.3 holey carbon Au 300 mesh grids, previously glow discharged and coated with 3 μl 0.2 mg/ml graphene oxide. The grids were blotted for 3.0 s at 100% humidity and 22 °C and plunged into liquid ethane using a Vitrobot Mark IV (Thermo Fisher Scientific).

To obtain dATP/ATP-loaded NrdR, 1 mM ATP, 1 mM dATP, and 10 mM MgCl$_2$ were added to apo-NrdR, the sample was concentrated to 3 mg/ml and loaded on a Superdex 200 as described above in a buffer containing 50 mM Tris-HCl pH 8.0, 150 mM NaCl, 10 mM MgCl$_2$, 1 mM TCEP, 0.4 mM ATP, and 0.4 mM dATP. About 0.2 ml fractions from the middle of the peak were collected. NrdR at a concentration of 0.4 mg/ml (corresponding to 19 μM monomeric NrdR) was used to prepare grids as described for ATP-loaded NrdR, but on a Quantifoil R2/1 holey carbon Au 300 mesh grid.

To obtain dATP/ATP-loaded NrdR bound to its cognate DNA, double-stranded (ds) oligonucleotides containing the *nrdRJ* promoter region (Supplementary Table 2) were annealed as described above to obtain a stock concentration of 22 μM. SEC-purified dATP/ATP-loaded NrdR was concentrated to 0.8 mg/ml. Equal volumes of NrdR and DNA were mixed, followed by a 1:1 dilution in double-distilled water and the addition of more ATP and dATP. The final sample contained 9.5 μM (monomeric) NrdR (0.2 mg/ml) and 5.5 μM DNA in 25 mM Tris-HCl pH 8.0, 75 mM NaCl, 5 mM MgCl$_2$, 0.7 mM ATP, 0.7 mM dATP, 0.5 mM TCEP. Grids were prepared as described for ATP-loaded NrdR.

Cryo-EM experiments were conducted at the Cryo-EM Swedish National Facility, SciLifeLab, Stockholm and Umeå nodes. Images for all samples were collected using EPU software on a Titan Krios Electron microscope (Thermo Fisher Scientific) operating at 300 kV, using a Falcon-III direct electron detector for ATP-loaded, dATP/ATP-loaded and DNA-bound dATP/ATP-loaded NrdR, and a Gatan K2 Summit direct detector coupled with a Bioquantum energy filter with 20 eV slit for dATP/ATP-loaded NrdR. The specific data collection and processing parameters for each sample are summarized in Supplementary Table 1. Patch motion correction, patch contrast transfer function (CTF) determination, particle picking, 2D and 3D classification, and refinement were performed using cryoSPARC version 3.1[42]. Initial particle picks were performed manually, and final picks were carried out using the cryoSPARC template-based picker with initial 2D

classes as input. After 2D classification and ab-initio map reconstruction, initial maps were refined using cryoSPARC nonuniform refinement, and local and global CTF refinement were performed after sorting particles into exposure groups. The NrdR models were built into the maps using Coot[43] and Chimera[44], together with real-space refinement in Phenix[45]. 3DFSC analyses[46] and model-to-map FSC curves calculated in Phenix (version 1.19.2-4158-000)[45] are shown in Supplementary Fig. 14. Structure and volume representation figures were prepared using PyMOL Molecular Graphics System and UCSF Chimera.

**Gas-phase electrophoretic mobility molecular analysis (GEMMA)**. In GEMMA, biomolecules are electrosprayed into the gas phase and neutralized to singly charged particles, and the gas-phase electrophoretic mobility is measured with a differential mobility analyzer. The mobility of an analyzed particle is proportional to its diameter, which therefore allows for quantitative analysis of the different particle sizes contained in a sample[47]. The GEMMA instrumental setup and general procedures were described previously[48]. NrdR and NrdRΔNΔC proteins were equilibrated by Sephadex G-25 chromatography into a buffer containing 100 mM NH$_4$CH$_3$CO$_2$, pH 7.3, and 0.5 mM TCEP. Prior to GEMMA analysis, the protein samples were diluted to a concentration of 0.08–0.25 mg/ml in a buffer containing 100 mM NH$_4$CH$_3$CO$_2$, pH 7.3, 0.005% (v/v) Tween-20, 50 μM nucleotide (when indicated), and Mg(CH$_3$CO$_2$)$_2$ equimolar to the total nucleotide concentration, incubated for 5 min at room temperature, centrifuged and applied to the GEMMA instrument. Concentrations of NrdR were 0.08 and 0.16 mg/ml (corresponding to 3.8 and 7.6 μM monomeric NrdR). The lower concentration was used for apo-NrdR, NrdR with dATP + ATP, ATP, dATP, ADP, dAMP, c-di-AMP. The higher concentration was used for NrdR samples with dATP + ATP, dATP + ADP, dADP, and ADP. Concentrations of NrdRΔNΔC was 0.08 mg/ml (6 μM) for samples with ATP + dATP, ATP or dATP, 0.12 mg/ml (9 μM) for samples with ATP or dATP, and 0.25 mg/ml (19 μM) for apo-NrdRΔNΔC. The oligomeric state of samples was not affected by different protein concentrations. The runs were conducted at a low flow rate, resulting in 1.4–2 p.s.i. pressure. The samples were scanned several times (3–10) to increase the signal-to-noise ratio and added together to obtain the traces presented here. The GEMMA system contained the following components: 3480 electrospray aerosol generator, 3080 electrostatic classifier, 3085 differential mobility analyzer, and 3025 A ultrafine condensation particle counter (TSI Corp., Shoreview, MN).

**Analytical size-exclusion chromatography**. The SEC experiments were performed at a temperature of 7 °C with a Superdex 200 PC 3.2/30 (with a total volume of 2.4 ml) and an ÄKTA prime system (Cytiva). The column was equilibrated with SEC buffer containing 50 mM Tris-HCl pH 8.5, 50 mM KCl, 10 mM MgCl$_2$, and 0.5 mM TCEP. The injection loop volume was 25 μl and the flow rate was 0.1 ml/min. To study nucleotide-dependent protein oligomerization, 64 μM NrdR samples were used. MgCl$_2$ was added to a final concentration of 10 and 1 mM of either ATP, dATP, ADP, dADP, AMP, dAMP, or their combinations were added, unless indicated differently. The samples were preincubated for 5 min at room temperature, centrifuged, and applied to the column. Nucleotides were also included in the SEC buffer to avoid dissociation of nucleotide-induced protein complexes during the run. Concentrations of nucleotides in the buffer were at least 100 times higher than the K$_D$s determined by ITC to ensure that the protein was occupied by the tested nucleotide during the entire SEC run; 0.2 mM for dATP; 0.5 mM for ATP, ADP, and dADP; 1 mM for AMP and dAMP. The following combinations of nucleotides were used: ATP and dATP at 0.25 mM each; ADP and dATP at 0.5 mM each; 3 mM ATP and 0.2 mM dATP. Molecular weights were estimated based on a calibration curve, derived from globular protein standards using a high molecular weight SEC marker kit (Cytiva) and may not reflect the actual mass of NrdR, due to the non-globular shape of the complexes.

**Reporting summary**. Further information on research design is available in the Nature Research Reporting Summary linked to this article.

## Data availability

The data that support this study are available from the corresponding authors upon reasonable request. Three-dimensional cryo-EM maps generated during this study have been deposited in the Electron Microscopy Data Bank (EMDB) under accession codes EMD-13178 (ATP-loaded NrdR dodecamer), EMD-13179 (dATP/ATP-loaded NrdR tetramer bound to its cognate DNA), and EMD-13182 (dATP/ATP-loaded NrdR octamer). Coordinates of all models have been deposited in the Protein Data Bank (PDB) under accession codes 7P37 (ATP-loaded NrdR dodecamer), 7P3F (dATP/ATP-loaded NrdR tetramer bound to its cognate DNA), and 7P3Q (dATP/ATP-loaded NrdR octamer). The Genome Taxonomy Database is available at (https://gtdb.ecogenomic.org). The RefSeq database is available at (https://www.ncbi.nlm.nih.gov/refseq/). Source data are provided with this paper.

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

## Acknowledgements

The authors thank Anders Hofer, Umeå University, for letting us use the GEMMA instrument, Ronnie Berntsson, Umeå University for help with the SEC-MALS analysis, Henrietta Nielsen, Stockholm University, for letting us use the MST instrument, and Malvern Panalytical for kindly sharing the MicroCal PEAQ-ITC analysis software for the analysis of ITC data. Cryo-EM sample screening, optimization, and data collection were performed at the Cryo-EM Swedish National Facility, funded by the Knut and Alice Wallenberg, Family Erling Persson and Kempe Foundations, SciLifeLab, Stockholm University and Umeå University. The authors would like to thank Michael Hall, Marta Carroni, Dustin Morado, Karin Wallden, and Julian Conrad for their assistance during the cryo-EM experiments. This work was supported by Swedish Research Council Grants 2018-03406 (to P.S.), 2016-04855 (to D.T.L.), and 2019-01400 (to B.-M.S.); the Swedish Cancer Foundation Grants 20 1287 PjF (to P.S.) and 2018/820 (to B.-M.S.), funds from the Wenner-Gren Foundation to B.-M.S., and S.S. was supported by a stipend from Carl Trygger Foundation to I.R.G. (grant CTS 20:361).

## Author contributions

I.R.G., D.T.L., B.-M.S., and P.S. conceived the project; I.R.G., S.S., and O.B. prepared protein and protein–DNA samples; I.R.G. and S.S. performed nucleotide- and DNA-binding, GEMMA and SEC analyses; G.N. and D.L. performed bioinformatic analyses; M.M.-C. performed cryo-EM sample preparation, data collection, and image analysis; M.M.-C. and P.S. constructed the atomic models with assistance from D.T.L.; B.-M.S. and P.S. supervised the project; All authors contributed to the preparation of the manuscript, discussed results, and approved the manuscript. I.R.G. and M.M.-C. contributed equally to this work.

## Funding

## Competing interests
The authors declare no competing interests.
