## [Peer Review File · Nature Communications]

A nucleotide-sensing oligomerization mechanism that controls NrdR-dependent transcription of ribonucleotide reductasesREVIEWER COMMENTS

Reviewer #1 (Remarks to the Author):

The authors in this manuscript reported three cryo-EM structures of NrdR, a transcription repressor of ribonucleotide reductases. They showed that NrdR exhibits distinct oligomerization states upon engagement of different combinations of nucleotide ligands. The key finding in this study is that NrdR directly responds to the cellular ATP/dATP ratio by exchanging the bound nucleotides in its ligand-binding pocket and adjusting its oligomerization states for DNA binding. This manuscript describes structural details of a new means of ligand-activated transcription in bacteria. I would like to recommend publication of this work with some revisions in terms of the following points.

1. In the structure of ATP-bound dodecamer (trimer of tetramers), are the monomers in the three tetramers organized in a similar manner? Please include supplementary figure panels for detailed comparison. Moreover, the authors mentioned that the two tetramers of the ATP/dATP-bound octamer are very similar to the tetramer that interacts with dsDNA. Please also include supplementary figure panels (or r.m.s.d values) to illustrate the similarity.
2. Although the authors described the changes of inner/outer pockets upon dATP binding, it is not clear in the current manuscript how the displacement of ATP by dATP in the outer pocket induces dissociation of dodecamer, and how the displacement of ATP by dATP in the outer pocket creates new interface for octamer. Please prepare additional figure panels and texts to describe the details.
3. The authors showed that the ATP/dATP-bound NrdR is in equilibrium between octamer and tetramer in solution. Was there a certain population of single particles of ATP/dATP-bound NrdR tetramer in the cryo-EM dataset of ATP/dATP-bound NrdR? How large is the interface of the two tetramers in the ATP/dATP-bound NrdR octamer? Why is the ATP/dATP-bound octamer prone to dissociation? In contrast, why is the ATP-bound dodecamer much more stable compared with the ATP/dATP-bound octamer?
4. Figure 3. Please label the core promoter region (-35 and -10 elements) and the putative transcription start site of the promoter sequence. Please prepare supplementary figure panels to show the cryo-EM map of the DNA-contacting residues to support the model.
5. Please prepare supplementary figure panels to show whether preferred orientations exist in the three cryo-EM datasets.

6. Fig.1, what are the criteria to determine the KDs '>' or '>>' 10 micromolar?

7. In the method section (p.13, l. 440-454), why were different concentrations of nucleotides used in the SEC buffer?

Reviewer #2 (Remarks to the Author):

Review report for “Evolutionarily mobile ATP-cones control both transcription and activity of an essential enzyme family”

Ribonucleotide reductase (RNR) is an essential enzyme that catalyzes the synthesis of DNA building blocks. NrdR, an RNR-specific repressor, controls its transcription in the majority of bacteria and some archaea. The authors present three cryo-EM structures of NrdR from *Streptomyces coelicolor* binding with ATP, ATP/dATP or ATP/dATP/DNA, and suggest a mechanism of different oligomer states of NrdR mediated by ATP or/and dATP, providing structural basis for NrdR binding to DNA and repressing transcription of RNR. The proposed mechanism revised the previously suggested hypotheses of NrdR functioning. The reported structures are significant improvement for NrdR regulating the transcription of RNRs by sensing concentrations of ATP and dATP. While the reported structures are interesting and contribute to the mechanistic study of evolutionarily mobile ATP-cone proteins, many concerns on the observed structures and proposed mechanisms need to be addressed before further consideration.

Major comments:

1. The observed ATP-loaded NrdR12 and dATP/ATP-loaded NrdR8 may be artifact oligomer states due to a high concentration of the protein used in grid preparation (19 μ M monomeric NrdR), which commonly causes the generation of higher oligomer states. Whether these states are biologically relevant needs further strong evidence. Similarly, the supportive analytical size exclusion chromatography analyses shown in Figure 4 may also result from the usage of an extremely high concentration of protein (64 μ M). Is this concentration close to the physiological one in bacterial cells? In the cryo-EM sample preparation for NrdR/ATP/dATP/DNA complex, why the final buffer condition is different from the other two, especially the NaCl concentration? Since it was reported that NaCl could significantly influence the different oligomer states of NrdR (Inna G, J Bacteriol. 2009, PMID: 19047342). Thus, the biological relevance of both NrdR12 and NrdR8 needs additional experiments to support.

2. For cryo-EM data processing (Extended Data Figure 5), the authors have omitted lots of important details. No detailed information about 2D classification and 3D classification was presented. Could the authors elaborate the data processing with more details in the corresponding Figure and Methods Sections? Only one 3D class for each of the three individual cryo-EM datasets was displayed in the figure. Are there any other different conformations of 3D classes for each dataset? What's the percentage of the presented 3D class? Those omitted classes may have different oligomer states. Without clear representations of these information, it's hard to believe that the proposed mechanism (Figure 2) would challenge/revise earlier proposed ones.

3. The resolutions are relatively good for the structures of ATP-loaded NrdR12 and dATP/ATP-loaded NrdR8. However, based on the presented map (Extended Data Figure 4C), it's hard to determine if dATP is the right choice for building. In the reviewer's opinion, ATP is more suitable since there clearly exists extra density for the hydroxyl (-OH) group in the 2' position. New figures with clear map and orientation showing a reliable building of dATP is necessary.

4. This study has proposed a revised action mechanism of NrdR that differs from the previously reported one. However, the authors have not discussed this point. Could the authors elaborate what the previous mechanism was proposed, discuss the differences between two distinct mechanisms and add a paragraph to explain it if possible.

5. The map quality of the most important result: how NrdR4 interacts with and bends DNA is poor. Based on the presented maps (Extended Data Figures 5 and 6), there may exist the preferred orientation issue for this cryo-EM dataset. Detailed 3D-Histogram and directional FSC plots for all the cryo-EM maps using 3DFSC web server should be presented to show the overall qualities (<https://www.nature.com/articles/nmeth.4347> and <https://3dfsc.salk.edu/>). Additionally, there is no clear figure showing the density of interactions between NrdR4 and DNA (Figure 3). If the local resolution is unable to unambiguously define the interactions, a clear statement such as "the current local resolution doesn't support confident side-chain assignment" should be included and the relevant descriptions on the interactions are expected to be tuned down. In addition, mutation analyses of the interface residues on the DNA-binding affinity are necessary for supporting the structural model.

Minor comments:

1. The current title "Evolutionarily mobile ATP-cones control both transcription and activity of an essential enzyme family" may not be suitable for the presented results because "The mobile ATP-cones control activity of the enzyme" is not the subject of this manuscript. This study only focused on how ATP-cones of NrdR sense ATP/dATP to change oligomer states and bind to DNA, therefore repressing the transcription of RNR.

2. In page 2 line 67, "ligate" should be replaced with "coordinate".

3. To better understand where NrdR binds in the promoter DNA, could the authors label and indicate the -10 and -35 element sequences on the DNA (Figure 3) used in cryo-EM study, since the double stranded (ds) oligonucleotides containing the nrdRJ promoter region.
4. In page 6 line 154, it seems to lack a word after “K62”.
5. When describing the interactions between ribose and these residues, please add a figure reference in page 6 line 163.
6. Indicate 6 Å (page 6 line 168) and 8 Å (page 6 line 171) distance movement using arrows in the corresponding figures if possible.
7. Page 6 line 173, there is no “K69” in figure 2C. If “K69” is only shown in Extended Data Figure, add the figure reference.
8. Page 7 line 182, in the absence of nucleotides NrdR elutes as a dimer, but not 100% (Figure 4). Some ratios of tetramers or octamers still exist. Could the authors explain this? Better to add “mainly” in the sentence.
9. Page 10 line 303 and page 11 line 347, put the exact concentrations before the compound names.
10. Page 12 line 419 and line 421, change “CryoSPARC” or “CryosPARC” to “cryoSPARC” and also add the version number.
11. An additional figure showing FSCs of model-to-map should be presented to evaluate the overall agreement of the experimental density map with a density map derived from the coordinate model (model map).

Reviewer #3 (Remarks to the Author):

This manuscript by Grinberg *et al.*, Sjöberg and Stenmark describes studies on NrdR, the transcription repressor of the ribonucleotide reductase (RNR) in *Streptomyces coelicolor*. The authors carry out a series of structural studies on NrdR in complex with ATP, ATP/dATP or ATP/dATP-cognate DNA. They also carry out a series of biochemistry studies on NrdR to determine the binding affinities of this repressor for a different nucleotides. Their bottom line conclusion is that previously proposed mechanisms of the “activation” of NrdR are incorrect and they provide a very interesting structure-based alternative for NrdR function in which namely an NrdR-dodecamer-(ATP)₂ complex cannot bind cognate DNA nor can an octameric NrdR-(dATP/ATP) intermediate complex but a tetrameric NrdR-(dATP/ATP) complex is DNA-binding competent and hence acts as a repressor, which likely occurs during stationary phase or when the cell is replete with dATP precluding the need for RNR activity. The structural biochemistry is very interesting and does indeed provide support for their proposed, alternative mechanism of how NrdR is regulated by nucleotides and functions. However,

there are several issues that are a bit confusing or seemingly incorrect that must be addressed as well as missing data.

The most fundamentally confusing issue is the data from their buried/supplemental ITC data (Extended Data Figure 8). The authors provide N values of typically 0.5. They do not explain what this N value means, which this reviewer thinks is 2 nucleotides per protomer. They then provide K_d for this binding event using a one-set-of sites binding model. This cannot be the correct method to fit these data, as their own structural data shows two independent binding sites. From their structures, these sites are not equivalent, i.e., they involve different residues of each protomer to bind. The K_d values should be different. Is there any cooperativity between the sites? Moreover, the authors should mutate key residues in each of the nucleotide binding pockets. (shown in Figures 2C,D and Extended Data Figure 4) to examine their effects on the binding. These data have to be re-examined and clarified.

Given the K_d values of the adenine-containing nucleotides, why does ADP bind more tightly than ATP?

The authors add 10 mM $MgCl_2$ to their cryo-EM samples, yet Mg^{+2} does not appear in their structures (it could not be found in the PDB validation reports). Is $MgCl_2$ required for ATP/dATP binding to NrdR? The assumption is yes it is necessary and if so, why is it not included/visualised in the current model.

Page 2, lines 53 -54: The authors state in the Introduction that “Several hypotheses on how NrdR regulates RNR expression have been proposed⁶⁻⁹...”, but never state what these are here. The reader finds out later on only one hypothesis. It is critical that these be summarised in the introduction and detailed better in the discussion.

Page 5, paragraph 1: The authors describe the interactions between the octameric NrdR-(ATP/dATP)-NrdR binding boxes. They state that residues D15 and R27 form an “intricate network with the exposed bases in the palindromic *NrdR* boxes”. They never state with which bases these residues interact. This is critical. Furthermore, from inspection of these residues in Figure 3B, the reader cannot fathom how they interact and how they provide specificity for the *NrdR* boxes. If these data are not already available, the authors must mutate these residues to test their importance not only in binding affinity but in specificity. What base(s) does the carboxylate side chain of residue D15 contact? If the authors know these, those bases should also be mutated and the affinities measured. These data might already be published or known and if so, must be included in this manuscript.

It is interesting that the conserved R₄ cluster and two other arginines are involved in the bending and distortion of the DNA. However, how do they contribute to NrdR binding to specific sites, as they make only phosphodiester backbone contacts. Should not they be able to do this with any DNA site in a “non sequence specific” manner?

Page 5: The authors fit the DNA density with their *nrdRJ* promoter sequence. How was this done? How do they know the register of the DNA given the medium resolution (3.31 Å)? The density of the DNA, at least gleaned from the video, is not that high resolution and hence, it is not clear how the authors would be able to fit it. If they are confident with the model building they should state this in the manuscript and how it was done, as this is critical.

Page 6: The authors state that mutation of residue V48 impairs nucleotide binding. Why? This residue is never shown in any figure. Please clarify.

Figure 2C and 2D should be made larger and the density should be removed. Residue Y128 might be better visualised as a CPK or sphere to highlight its different positions.

Figure 3C: It is not clear what is “yellow on black” versus “white on black”.

What prohibits other nucleotides from binding either the inner site or the outer site? Is the inner site always occupied by ATP, which seems reasonable? What about the outer site? What keeps the other nucleotides from binding, even if poorly relative to dATP or dADP or ATP? The authors should expand upon this. The authors should also state what specifies adenine ring recognition. From Extended Data Figure 4B, 4C, it would appear that residue E56 is critical. Is this correct and if not what residues are important for adenine ring specificity?

The abstract should be rewritten slightly. The wording “...the ATP-cone moonlights by both immediate inhibition of RNR enzyme activity and a long-lasting repression of RNR expression.” This implies that the ATP-cone of NrdR does both of these functions.

A figure of the density of the Zn atoms of the Zn ribbon would be a good addition as it would highlight the quality of the structures.

Reviewer #4 (Remarks to the Author):

The manuscript by Grinberg et al. describes the biochemical and structural analyses of the *S. coelicolor* NrdR. NrdR is a transcriptional regulator with an N-terminal Zn-ribbon DNA binding domain and a C-terminal ATP-cone domain. It uses its ATP-cone domain to sense nucleotide concentrations and represses transcription of ribonucleotide reductase (RNR). RNR is an essential enzyme in DNA synthesis as it generates deoxyribonucleotides from ribonucleotides. There are currently no structures of an NrdR bound to DNA and the effector nucleotides (ATP, AMP etc.), which regulate its activity have been the subject of debate. Here the authors used biochemical studies to determine that both dATP and ATP are required to activate the specific DNA binding function of NrdR. They then solve the Nrd-ATP complex, the NrdR complex with ATP and dATP and the NrdR-(dATP/ATP)DNA structures by cryo-EM. The latter represents the first structure of an NrdR protein bound to DNA. Not only is this the first NrdR-DNA structure but the authors also clear up/revise earlier hypothesis on the system and will be an important contribution to the field. But I have several questions/ concerns that the authors should address before publication.

1. The ITC data are a bit confusing to me; maybe this just needs better explanation. But these data were fit to a one-set-of-sites binding mode; that implies that the inner and outer nucleotide binding sites bind dATP (and ATP) with identical affinities, the same thermodynamics and the same mode. But these binding sites look quite different.
2. Since dATP and ATP appear to bind with the same stoichiometry- do these data then mean that dATP can also bind the inner sites?
3. If both sites bind with the same affinity to dATP and ATP then why, in the presence of both does dATP bind the outer sites and ATP the inner sites?
4. the authors say that D15 and R17 mediate contacts that are base specific. Either the authors should show the contacts that are specific or they need to tone down the discussion on the base specificity mediated by these residues. The resolution around the DNA also appears somewhat limited. If there are specific contacts these should be shown with density included.
5. For the cryo-EM structures. the authors don't show angular distribution so it is unclear if the reconstructions suffer from orientation bias. These data should be included.

6. The authors don't comment on whether their cryo-EM data for each structure had different populations of structures/conformations. In other words, were the particles all homogeneous in each structure or was there evidence for other species? In the NrdR(dATP-ATP)-DNA analyses were all the particles in the DNA bound form and all tetramers. Did this sample have any octameric, non-DNA bound particles?

7. Their NrdR(dATP-ATP) structure is octameric and is not compatible with DNA binding. The protein needs to dissociate to a tetramer to bind DNA. How is this accomplished? Is the equilibrium driven to a tetramer in the presence of DNA? The authors should discuss/address this question. Along the question above- what are the buried surface areas in the various interfaces?

8. As the NrdR(dATP-ATP)-DNA complex is tetrameric what happens to the other two DNA binding domains within the tetramer? Do they contact any part of the DNA?

If not, is the tetramer actually needed for DNA binding or could a dimer bind?

9. It would seem that dATP + ADP complex would bind DNA with higher affinity as this complex seems to be present as primarily the tetramer (Fig. 4). Can the authors comment on this.

10. Figure 2 is a key figure. But I found Figure 2A unclear in showing why the octamer cannot bind DNA. Also from the DNA bound complex it is unclear how the tetrameric subunits are associated. It would be helpful if the authors could provide clearer figures here even if it means including density maps and ribbon diagrams of each complex separately.

Response to reviewers

We thank the reviewers for their positive response and constructive suggestions. We have in principle followed all suggestions, and believe the reviewing process has significantly improved our manuscript. In particular the additional mutational and binding affinity studies of both NrdR and the recognition sites on the promoter DNA, that we performed in response to the reviewer's questions, have strengthened the conclusions drawn from our structural studies. Some editing changes have also been introduced to increase comprehensibility. Detailed responses to the reviewers' comments are included below (in blue). Additionally, a copy of the manuscript with marked changes is provided. Hopefully you will now find our manuscript acceptable for publication.

Reviewer #1 (Remarks to the Author):

The authors in this manuscript reported three cryo-EM structures of NrdR, a transcription repressor of ribonucleotide reductases. They showed that NrdR exhibits distinct oligomerization states upon engagement of different combinations of nucleotide ligands. The key finding in this study is that NrdR directly responds to the cellular ATP/dATP ratio by exchanging the bound nucleotides in its ligand-binding pocket and adjusting its oligomerization states for DNA binding. This manuscript describes structural details of a new means of ligand-activated transcription in bacteria. I would like to recommend publication of this work with some revisions in terms of the following points.

1. In the structure of ATP-bound dodecamer (trimer of tetramers), are the monomers in the three tetramers organized in a similar manner? Please include supplementary figure panels for detailed comparison. Moreover, the authors mentioned that the two tetramers of the ATP/dATP-bound octamer are very similar to the tetramer that interacts with dsDNA. Please also include supplementary figure panels (or r.m.s.d values) to illustrate the similarity.
Yes, the arrangements of the monomers in the three tetramers that build up the dodecamer are highly similar. D3 symmetry was used for the dodecamer dataset; this gives six symmetry copies of the A and B chains (Supplementary Table 1). Supplementary figure 8 presents an overlay of the chain A and chain B monomers from all three structures (dodecamer, octamer, tetramer). Supplementary figure 8 shows the high similarity of the tetrameric structure that interacts with dsDNA and octameric structures, and also the differences in the orientation of the zinc-ribbon domain between the A and B chains.
2. Although the authors described the changes of inner/outer pockets upon dATP binding, it is not clear in the current manuscript how the displacement of ATP by dATP in the outer pocket induces dissociation of dodecamer, and how the displacement of ATP by dATP in the outer pocket creates new interface for octamer. Please prepare additional figure panels and texts to describe the details.
A new figure 5 and supplementary video 4 and 5 have been added to the manuscript. An additional paragraph has been added to the manuscript at the end of the section "Nucleotide coordination differences between assemblies" further describing this complex structural rearrangement.
3. The authors showed that the ATP/dATP-bound NrdR is in equilibrium between octamer and tetramer in solution. Was there a certain population of single particles of ATP/dATP-bound NrdR tetramer in the cryo-EM dataset of ATP/dATP-bound NrdR? How large is the interface of the two tetramers in the ATP/dATP-bound NrdR octamer? Why is the

ATP/dATP-bound octamer prone to dissociation? In contrast, why is the ATP-bound dodecamer much more stable compared with the ATP/dATP-bound octamer?

Supplementary figure 6 shows the 2D alignments and particle orientation distributions for all three structures. The particle populations were quite homogeneous in all three cases, but in the early 2D alignments of the dataset that yielded the dATP/ATP-loaded NrdR octamer structure (Supplementary Fig. 6B) a small amount of tetrameric particles could be discerned. The relative count of these particles was low compared to the octameric particles.

Regarding dissociation of the octamer, the buried solvent accessible area in the interface between two tetramers is 3708 Å², compared to 8546 Å² buried within each tetramer, implying that interactions between tetramers are significantly weaker than those within tetramers. A similar analysis shows that the buried solvent accessible area in the interface between any pair of tetramers in the dodecamer is 4924 Å², indicating that these interactions are harder to break and that the dodecamer is more stable than the octamer. Text has been added to the sections describing the dodecamer and octamer.

4. Figure 3. Please label the core promoter region (-35 and -10 elements) and the putative transcription start site of the promoter sequence.

Such a figure has been prepared and shows the core promoter regions for the two RNR operons in *Streptomyces coelicolor*, with -35, -10, and transcription starts sites annotated. It is included as figure 1C.

Please prepare supplementary figure panels to show the cryo-EM map of the DNA-contacting residues to support the model.

Specific interactions between DNA and protein suggested in the cryo-EM structure are now supported by mutations in the DNA fragments as well as NrdR mutations (Fig. 4).

5. Please prepare supplementary figure panels to show whether preferred orientations exist in the three cryo-EM datasets.

Supplementary figure 6 shows the 2D alignments and particle orientation distributions for all three structures. The particle populations were quite homogeneous in all three cases and there are no major issues with preferred particle orientations (Supplementary Fig. 6). Also see the reply to Reviewer 2, question 5.

6. Fig.1, what are the criteria to determine the K_Ds '>' or '>>' 10 micromolar?

For some very weak interactions the curves do not reach a distinct plateau. In order to not overestimate such apparent K_Ds calculated by the analysis software, we denote the K_Ds between 10 μM and 19 μM as K_D >10 μM, and above 19 μM as K_D >>10μM. Relevant parts of this information is now included in legends to figures 1 and 4, and Supplementary figures 2, 3 and 9.

7. In the method section (p.13, l. 440-454), why were different concentrations of nucleotides used in the SEC buffer?

Different concentrations were used, based on K_D values determined by ITC, to ensure that the protein was occupied by the tested nucleotide during the entire SEC run. This explanation has been included in the Methods section.

Reviewer #2 (Remarks to the Author):

Review report for “Evolutionarily mobile ATP-cones control both transcription and activity of an essential enzyme family” Ribonucleotide reductase (RNR) is an essential enzyme that

catalyzes the synthesis of DNA building blocks. NrdR, an RNR-specific repressor, controls its transcription in the majority of bacteria and some archaea. The authors present three cryo-EM structures of NrdR from *Streptomyces coelicolor* binding with ATP, ATP/dATP or ATP/dATP/DNA, and suggest a mechanism of different oligomer states of NrdR mediated by ATP or/and dATP, providing structural basis for NrdR binding to DNA and repressing transcription of RNR. The proposed mechanism revised the previously suggested hypotheses of NrdR functioning. The reported structures are significant improvement for NrdR regulating the transcription of RNRs by sensing concentrations of ATP and dATP. While the reported structures are interesting and contribute to the mechanistic study of evolutionarily mobile ATP-cone proteins, many concerns on the observed structures and proposed mechanisms need to be addressed before further consideration.

Major comments:

1. The observed ATP-loaded NrdR₁₂ and dATP/ATP-loaded NrdR₈ may be artifact oligomer states due to a high concentration of the protein used in grid preparation (19 μ M monomeric NrdR), which commonly causes the generation of higher oligomer states. Whether these states are biologically relevant needs further strong evidence. Similarly, the supportive analytical size exclusion chromatography analyses shown in Figure 4 may also result from the usage of an extremely high concentration of protein (64 μ M). Is this concentration close to the physiological one in bacterial cells?

The oligomeric states of NrdR assayed with GEMMA are consistent with both the SEC and the cryo-EM results. In the GEMMA analyses protein concentrations were as low as 0.08 mg/ml, corresponding to 3.8 μ M monomeric protein (Fig. 6 and Supplementary Fig. 11). This information has been added to the Methods section.

In the cryo-EM sample preparation for NrdR/ATP/dATP/DNA complex, why the final buffer condition is different from the other two, especially the NaCl concentration?

Since it was reported that NaCl could significantly influence the different oligomer states of NrdR (Inna G, J Bacteriol. 2009, PMID: 19047342). Thus, the biological relevance of both NrdR₁₂ and NrdR₈ needs additional experiments to support.

The concentration of NaCl was decreased in the DNA-containing cryo-EM sample to promote better contrast in the cryo-EM micrographs and sustain tight binding. Interactions between proteins and DNA are primarily electrostatic, and lower salt concentrations would prevent dissociation of the protein-DNA complex, which can sometimes happen during grid freezing. Since the major form of NrdR oligomer in these grids is tetramer (the lowest oligomeric form observed for NrdR loaded with nucleotides), low salt is conceivably not inducing higher order oligomerization of NrdR. In addition, our GEMMA results using the same protein concentration (0.08 mg/ml, 3.8 μ M) in the absence and the presence of different nucleotide effectors show that the effector nucleotide determines the oligomeric state (Fig 6, Supplementary Fig. 11A and B). We assayed some samples at two different protein concentrations (3.8 μ M and 7.6 μ M for NrdR; 6 μ M and 9 μ M for NrdR Δ N Δ C) and showed that both concentrations resulted in similar elution profiles. This information has been added to the Methods section.

2. For cryo-EM data processing (Extended Data Figure 5), the authors have omitted lots of important details. No detailed information about 2D classification and 3D classification was presented. Could the authors elaborate the data processing with more details in the corresponding Figure and Methods Sections? Only one 3D class for each of the three individual cryo-EM datasets was displayed in the figure. Are there any other different conformations of 3D classes for each dataset? What's the percentage of the presented 3D

class? Those omitted classes may have different oligomer states. Without clear representations of these information, it's hard to believe that the proposed mechanism (Figure 2) would challenge/revise earlier proposed ones.

Supplementary figure 6 now shows 2D classes for all 3 structures, explaining that there was one main population in all cases. A small population of tetrameric NrdR particles was present in early 2D classifications of the dataset that yielded the octameric NrdR structure.

3D classification attempts did not reveal clear distinct 3D classes within the dodecameric, octameric or tetrameric particle sets.

3. The resolutions are relatively good for the structures of ATP-loaded NrdR12 and dATP/ATP-loaded NrdR8. However, based on the presented map (Extended Data Figure 4C), it's hard to determine if dATP is the right choice for building. In the reviewer's opinion, ATP is more suitable since there clearly exists extra density for the hydroxyl (-OH) group in the 2' position. New figures with clear map and orientation showing a reliable building of dATP is necessary.

Supplementary Figure 4C shows one ATP and one dATP modeled in the tetrameric NrdR bound to DNA. In our opinion there is clear density for a hydroxyl group in the 2' position of the ATP (at the end of the arrow extending from the ATP text in Supplementary figure 4C). However, there is no clear density for a hydroxyl group in the 2' position of the dATP. (Supplementary Figs. 4C and 4E). More importantly, there is no room for a hydroxyl group in the 2' position, it would clash with F124 (Supplementary Fig. 4E). We agree that the resolution is not sufficient to confidently build the nucleotides without the additional information from biochemical and biophysical considerations. We show that the combination of ATP and dATP is necessary for NrdR to bind DNA and that it is two distinct nucleotide binding sites (Fig. 1B, 2C, 5, Supplementary Figs. 2, 3, 4B, and answer to reviewer 4 question 3). It is therefore expected that the tetrameric NrdR bound to DNA has one ATP and one dATP bound. When looking at both nucleotide positions, with the prerequisite that one of them is ATP and one is dATP, the placement is clear. The opposite placement of ATP vs dATP would lead to disagreements with the density and to steric clashes

4. This study has proposed a revised action mechanism of NrdR that differs from the previously reported one. However, the authors have not discussed this point. Could the authors elaborate what the previous mechanism was proposed, discuss the differences between two distinct mechanisms and add a paragraph to explain it if possible.

The previously suggested mechanisms have been summarized in sections "DNA binding requires both dATP and ATP" and "Conclusions", where a comparison with the current mechanism deduced from our study is presented.

5. The map quality of the most important result: how NrDR4 interacts with and bends DNA is poor. Based on the presented maps (Extended Data Figures 5 and 6), there may exist the preferred orientation issue for this cryo-EM dataset.

Detailed 3D-Histogram and directional FSC plots for all the cryo-EM maps using 3DFSC web server should be presented to show the overall qualities

(<https://www.nature.com/articles/nmeth.4347> and <https://3dfsc.salk.edu/>). Additionally, there is no clear figure showing the density of interactions between NrDR4 and DNA (Figure 3). If the local resolution is unable to unambiguously define the interactions, a clear statement such as "*the current local resolution doesn't support confident side-chain assignment*" should be included and the relevant descriptions on the interactions are expected to be tuned down.

There is no major problem with preferred particle orientations in the NrDR4 dataset (Supplementary Fig. 6). There is clear density for most of the arginines that bind DNA.

However, the limited local resolution results in weak density for the side chains of some of the interacting amino acids. We are confident in the side-chain assignments in the interface. However, the precise orientation of the side chains of some amino acids is somewhat ambiguous because of low local resolution. A statement to this fact has been added to the manuscript. We have also toned down the detailed description of the interaction. As suggested by the reviewer 3DFSC results for all three structures have been added as Supplementary Figure 14.

In addition, mutation analyses of the interface residues on the DNA-binding affinity are necessary for supporting the structural model.

We have added the Results section “Mutations supporting DNA-protein interactions” where DNA-protein binding data are presented for single and double NrdR mutations as well as for several conserved base pair mutations in the NrdR boxes (Fig. 4, Supplementary Figs. 9-10). These results support DNA-protein interactions suggested in the cryo-EM structure.

Minor comments:

1. The current title “Evolutionarily mobile ATP-cones control both transcription and activity of an essential enzyme family” may not be suitable for the presented results because “The mobile ATP-cones control activity of the enzyme” is not the subject of this manuscript. This study only focused on how ATP-cones of NrdR sense ATP/dATP to change oligomer states and bind to DNA, therefore repressing the transcription of RNR. (Title has been changed).
2. In page 2 line 67, “ligate” should be replaced with “coordinate”. (fixed)
3. To better understand where NrdR binds in the promoter DNA, could the authors label and indicate the -10 and -35 element sequences on the DNA (Figure 3) used in cryo-EM study, since the double stranded (ds) oligonucleotides containing the nrdRJ promoter region. (fixed; Fig. 1C)
4. In page 6 line 154, it seems to lack a word after “K62”. (fixed)
5. When describing the interactions between ribose and these residues, please add a figure reference in page 6 line 163. (fixed)
6. Indicate 6 \AA (page 6 line 168) and 8 \AA (page 6 line 171) distance movement using arrows in the corresponding figures if possible. These distances are now shown in figure 5C and in Supplementary Video 4 and 5.
7. Page 6 line 173, there is no “K69” in figure 2C. If “K69” is only shown in Extended Data Figure, add the figure reference. (K69 is now shown in figure 2C.)
8. Page 7 line 182, in the absence of nucleotides NrdR elutes as a dimer, but not 100% (Figure 4). Some ratios of tetramers or octamers still exist. Could the authors explain this? Better to add “mainly” in the sentence (“mainly” added).
9. Page 10 line 303 and page 11 line 347, put the exact concentrations before the compound names. (the exact concentrations have been added)
10. Page 12 line 419 and line 421, change “CryoSPARC” or “CryosPARC” to “cryoSPARC” and also add the version number. (this has been corrected)
11. An additional figure showing FSCs of model-to-map should be presented to evaluate the overall agreement of the experimental density map with a density map derived from the coordinate model (model map). (Added model-to-map FSC curves calculated in Phenix in the right-side column of Supplementary Fig. 14.)

Reviewer #3 (Remarks to the Author):

This manuscript by Grinberg et al., Sjöberg and Stenmark describes studies on NrdR, the transcription repressor of the ribonucleotide reductase (RNR) in *Streptomyces coelicolor*. The

authors carry out a series of structural studies on NrdR in complex with ATP, ATP/dATP or ATP/dATP-cognate DNA. They also carry out a series of biochemistry studies on NrdR to determine the binding affinities of this repressor for a different nucleotides. Their bottom line conclusion is that previously proposed mechanisms of the “activation” of NrdR are incorrect and they provide a very interesting structure-based alternative for NrdR function in which namely an NrdR-dodecamer-(ATP)₂ complex cannot bind cognate DNA nor can an octameric NrdR-(dATP/ATP) intermediate complex but a tetrameric NrdR-(dATP/ATP) complex is DNA-binding competent and hence acts as a repressor, which likely occurs during stationary phase or when the cell is replete with dATP precluding the need for RNR activity. The structural biochemistry is very interesting and does indeed provide support for their proposed, alternative mechanism of how NrdR is regulated by nucleotides and functions. However, there are several issues that are a bit confusing or seemingly incorrect that must be addressed as well as missing data.

The most fundamentally confusing issue is the data from their buried/supplemental ITC data (Extended Data Figure 8). The authors provide N values of typically 0.5. They do not explain what this N value means, which this reviewer thinks is 2 nucleotides per protomer. They then provide K_d for this binding event using a one-set-of sites binding model. This cannot be the correct method to fit these data, as their own structural data shows two independent binding sites. From their structures, these sites are not equivalent, i.e., they involve different residues of each protomer to bind. The K_d values should be different. Is there any cooperativity between the sites? Moreover, the authors should mutate key residues in each of the nucleotide binding pockets. (shown in Figures 2C,D and Extended Data Figure 4) to examine their effects on the binding. These data have to be re-examined and clarified.

(Extended Data Fig. 8 is now called Supplementary Fig. 12) Unlike other parameters in ITC analyses, the N value is usually the least accurate of all fitted parameters and strongly dependent on the concentration of the active protein in the sample (Rozman Grinberg & al 2018a, 2018b, McKethan and Spiro 2013). N values described in the current study are in the range of 0.46 – 0.8 for different nucleotides, and are somewhat higher when experiments are performed at low temperature (10 °C). We therefore interpret the data as 1 high affinity-binding site per ATP-cone for each nucleotide, i.e. in the case of ATP binding we only detect one high affinity-binding site per ATP-cone. Our cryo-EM data shows that ATP binds with high occupancy to the inner site and with lower occupancy to the outer site in chain A and C (Supplementary Fig. 7C). We can however not detect/distinguish this low affinity binding of ATP to the outer site using ITC. Regarding mutation of residues in the binding pockets, a thorough description of earlier mutational studies of the ATP-cones in *S. coelicolor* and *E. coli* NrdR proteins has been added to the Results section “Nucleotide coordination differences between assemblies”.

Given the K_d values of the adenine-containing nucleotides, why does ADP bind more tightly than ATP?

The structure of the dodecamer that forms in the presence of ATP shows that the γ -phosphate groups of both the inner and outer ATP molecules make multiple interactions with side chains in the protein. In the ITC measurements, considering the series ATP, ADP and AMP, ΔH decreases from -112 to -81 to -32 kJ/mol, presumably because of a decreased number of interactions with the protein, but $-\Delta S$ also decreases from 82 to 46 to 2 kJ/mol, presumably due to decreased loss of conformational entropy and a decreased desolvation penalty upon nucleotide binding. The effects are compensatory, resulting in minor differences in ΔG of only a few kJ/mol (-30, -35, -31 kJ/mol) and differences in K_d of 4- to 5-fold. The energetic differences can not be rationalized completely in terms of structure, particularly since a large

number of protein-protein interactions, mostly between ATP-cones, are also generated upon nucleotide binding, but the structure and ITC data together support the idea of enthalpy-entropy compensation.

The authors add 10 mM MgCl₂ to their cryo-EM samples, yet Mg²⁺ does not appear in their structures (it could not be found in the PDB validation reports). Is MgCl₂ required for ATP/dATP binding to NrdR?

Nucleotide binding to NrdR requires MgCl₂, and ITC experiments in the absence of MgCl₂ results in aberrant binding (Rozman Grinberg, unpublished). Nucleotides in cells are bound by Mg ions, and physiological Mg²⁺ concentrations for *E. coli* are 1-2 mM free, 20-100 mM total (Theillet FX et al., Chem Rev. 2014).

The assumption is yes it is necessary and if so, why is it not included/visualised in the current model.

The resolution of the cryo-EM maps does not support the placement of the Mg²⁺ ion with confidence, although we believe the ion is indeed coordinated between the phosphate groups of the nucleotides. We now discuss this in the section “Nucleotide coordination differs between assemblies”.

Page 2, lines 53 -54: The authors state in the Introduction that “Several hypotheses on how NrdR regulates RNR expression have been proposed 6-9...”, but never state what these are here. The reader finds out later on only one hypothesis. It is critical that these be summarised in the introduction and detailed better in the discussion.

The previously suggested mechanisms have been described in Results sections “DNA binding requires both dATP and ATP” and “Conclusions”, where a comparison with the current mechanism deduced in our study is presented.

Page 5, paragraph 1: The authors describe the interactions between the octameric NrdR-(ATP/dATP)-NrdR binding boxes. They state that residues D15 and R27 form an “intricate network with the exposed bases in the palindromic NrdR boxes”. They never state with which bases these residues interact. This is critical. Furthermore, from inspection of these residues in Figure 3B, the reader cannot fathom how they interact and how they provide specificity for the NrdR boxes. If these data are not already available, the authors must mutate these residues to test their importance not only in binding affinity but in specificity. What base(s) does the carboxylate side chain of residue D15 contact? If the authors know these, those bases should also be mutated and the affinities measured. These data might already be published or known and if so, must be included in this manuscript.

We have added the Results section “Mutations supporting DNA-protein interactions” where DNA-protein binding data are presented for single and double NrdR mutations as well as for several conserved base pair mutations in the NrdR boxes (Fig. 4, Supplementary Figs. 9-10). These results support DNA-protein interactions suggested in the cryo-EM structure.

It is interesting that the conserved R4 cluster and two other arginines are involved in the bending and distortion of the DNA. However, how do they contribute to NrdR binding to specific sites, as they make only phosphodiester backbone contacts. Should not they be able to do this with any DNA site in a “non sequence specific” manner?

As the reviewer correctly points out, these arginines interact with the phosphodiester backbone and do not contribute to the specificity by direct interactions with the bases. The NrdR box has a central, weakly stacking, AT base pair. The NrdR boxes are therefore predispositioned for facilitating the 90° kink observed in the structure. The amino acids that

directly interact with the bases provide specificity also by disrupting the canonical base pairing, flipping out bases and therefore further facilitating the bending of the DNA. The distances between the conserved arginines that bind the backbone of the DNA together with the interaction that disrupt the canonical base pairing to directly interact with bases (e.g. D15 and R17) and the weak stacking of particular base pairs, all work together to allow the specific bending and recognition of the DNA in the NrdR recognition site. In addition to our own mutational studies of conserved base pairs and conserved protein residues, a mutational study of the R4 cluster in *E. coli* NrdR, has been described in the Results section “Mutations supporting DNA-protein interactions”.

Page 5: The authors fit the DNA density with their nrdRJ promoter sequence. How was this done? How do they know the register of the DNA given the medium resolution (3.31 Å)? The density of the DNA, at least gleaned from the video, is not that high resolution and hence, it is not clear how the authors would be able to fit it. If they are confident with the model building they should state this in the manuscript and how it was done, as this is critical. At this resolution we can not confidently assign the identity of the individual bases in the DNA, when looking at them individually. However, we can distinguish the general structure of almost the entire DNA construct we used (50 of 57 bases). This together with the two-fold symmetry in the DNA and in the NrdR tetramer puts restraints on the registry of the DNA, in relation to the NrdR tetramer. These structural considerations in combination with the mutational analysis of both the protein and the DNA constructs, that we now have performed, make us confident in the positioning of the DNA. Text describing the fitting of the DNA has been added to the manuscript.

Page 6: The authors state that mutation of residue V48 impairs nucleotide binding. Why? This residue is never shown in any figure. Please clarify. A thorough description of earlier mutational studies of the ATP-cones in *S. coelicolor* and *E. coli* NrdR proteins have been added to the Results section “Nucleotide coordination differences between assemblies”. The structures show that V48 is directly facing the base of the ATP in the inner binding site. V48 is located on a key β -hairpin illustrated in new figure 5A and 5C.

Figure 2C and 2D should be made larger and the density should be removed. Residue Y128 might be better visualised as a CPK or sphere to highlight its different positions. The density has been removed. Y128 is now highlighted in ball-and-stick.

Figure 3C: It is not clear what is “yellow on black” versus “white on black”. Yellow on black changed to orange on black.

What prohibits other nucleotides from binding either the inner site or the outer site? Is the inner site always occupied by ATP, which seems reasonable? What about the outer site? What keeps the other nucleotides from binding, even if poorly relative to dATP or dADP or ATP? The authors should expand upon this. The authors should also state what specifies adenine ring recognition. From Extended Data Figure 4B, 4C, it would appear that residue E56 is critical. Is this correct and if not what residues are important for adenine ring specificity?

The inner site is always occupied by ATP while the outer site can be occupied either by ATP or by dATP. Regarding the identity of the inner site nucleotide base, it has to be a purine to fill the pocket. The adenine base is recognized by two hydrogen bonds from the main chain of the loop between the β -hairpin and the first helix of the ATP-cone motif that match the 6-

NH₂ group and the unprotonated ring N1 atom in adenosine. There is also an H-bond from E56 to the 6-NH₂ group. Replacing adenosine by guanosine would result in three H-bonding mismatches and also a steric clash with F58 due to the 2-NH₂ group. For the outer site of the octamer, the 6-NH₂ group of adenosine makes an H-bond to the carbonyl group of K69 in the first helix of a neighboring monomer, which is again only possible for adenosine. In the octamer there may even be an H-bond between 6-NH₂ of ATP in one monomer and N1 of ATP in the neighboring one, while in the dodecamer the two adenosine rings stack on each other. We have added text to this effect under the section “Nucleotide coordination differences between assemblies”.

The abstract should be rewritten slightly. The wording “...the ATP-cone moonlights by both immediate inhibition of RNR enzyme activity and a long-lasting repression of RNR expression.” This implies that the ATP-cone of NrdR does both of these functions. The Abstract has been modified as suggested.

A figure of the density of the Zn atoms of the Zn ribbon would be a good addition as it would highlight the quality of the structures.

The map does not support the placement of the Zn atom with high precision, but the four cysteines coordinating the Zn atom can be accurately modeled in. The conformation of the cysteines would not be possible without the Zn atom positioned as in the models. Additionally, the Zn sites are located in the outside of the structures, where the local resolution is worse than the average, and might not be a good indicator of the overall resolution of the maps. An earlier study showed that the mutation C3A in *S. coelicolor* NrdR drastically reduces the amount of zinc bound and abolishes binding to the *nrdAB* and *nrdRJ* promoters (Grinberg & al 2006). This study also reported 0.7-0.8 Zn bound per monomer NrdR, as we have confirmed by total-reflection X-ray fluorescence (TXRF) studies in the current study (section Protein purification).

Reviewer #4 (Remarks to the Author):

The manuscript by Grinberg et al. describes the biochemical and structural analyses of the *S. coelicolor* NrdR. NrdR is a transcriptional regulator with an N-terminal Zn-ribbon DNA binding domain and a C-terminal ATP-cone domain. It uses its ATP-cone domain to sense nucleotide concentrations and represses transcription of ribonucleotide reductase (RNR). RNR is an essential enzyme in DNA synthesis as it generates deoxyribonucleotides from ribonucleotides. There are currently no structures of an NrdR bound to DNA and the effector nucleotides (ATP, AMP etc.), which regulate its activity have been the subject of debate. Here the authors used biochemical studies to determine that both dATP and ATP are required to activate the specific DNA binding function of NrdR. They then solve the Nrd-ATP complex, the NrdR complex with ATP and dATP and the NrdR-(dATP/ATP)DNA structures by cryo-EM. The latter represents the first structure of an NrdR protein bound to DNA. Not only is this the first NrdR-DNA structure but the authors also clear up/revise earlier hypothesis on the system and will be an important contribution to the field. But I have several questions/ concerns that the authors should address before publication.

1. The ITC data are a bit confusing to me; maybe this just needs better explanation. But these data were fit to a one-set-of-sites binding mode; that implies that the inner and outer nucleotide binding sites bind dATP (and ATP) with identical affinities, the same thermodynamics and the same mode. But these binding sites look quite different.

N values described in the current study are in the range of 0.46 – 0.8 for different nucleotides, and are somewhat higher when experiments are performed at low temperature (10 °C). We therefore interpret the data as 1 high affinity-binding site per ATP-cone for each nucleotide, i.e. in the case of ATP binding we only detect one high affinity-binding site per ATP-cone. Our cryo-EM data shows that ATP can bind with high occupancy to the inner site, and also to the outer site with lower occupancy. We therefore assume that the ITC detects one ATP binding to the inner site and one dATP binding to the outer site. The K_{DS} for ATP and dATP are not the same; ATP has 4-5 times lower affinity for the inner site than dATP has for the outer site.

2. Since dATP and ATP appear to bind with the same stoichiometry- do these data then mean that dATP can also bind the inner sites?

We have not detected binding of dATP to the inner site in this study, but an earlier study on mutations in *S. coelicolor* NrdR reported copurification of 2 dATP molecules with the Y128A mutant protein (Grinberg & al 2009).

3. If both sites bind with the same affinity to dATP and ATP then why, in the presence of both does dATP bind the outer sites and ATP the inner sites?

The K_{DS} for ATP and dATP are not the same; ATP has 4-5 times lower affinity to the inner site than dATP has for the outer site. We interpret the data as 1 high affinity-binding site per ATP-cone for each nucleotide, i.e. in the case of ATP binding we only detect one high affinity-binding site per ATP-cone. Our cryo-EM data shows that ATP can also bind to the outer site (Supplementary Fig. 7C). We can however not detect/distinguish this low affinity ATP binding using ITC.

4. the authors say that D15 and R17 mediate contacts that are base specific. Either the authors should show the contacts that are specific or they need to tone down the discussion on the base specificity mediated by these residues. The resolution around the DNA also appears somewhat limited. If there are specific contacts these should be shown with density included. We have added the Results section “Mutations supporting DNA-protein interactions” where DNA-protein binding data are presented for single and double NrdR mutations as well as for several conserved base pair mutations in the NrdR boxes (Fig. 4, Supplementary Figs. 9-10). These results support DNA-protein interactions suggested from the cryo-EM structure.

5. For the cryo-EM structures. the authors don't show angular distribution so it is unclear if the reconstructions suffer from orientation bias. These data should be included.

The data does not suffer from any major orientation bias. Supplementary figure 6 has been added, which shows the particle orientation distribution for all three structures.

6. The authors don't comment on whether their cryo-EM data for each structure had different populations of structures/conformations. In other words, were the particles all homogeneous in each structure or was there evidence for other species? In the NrdR(dATP-ATP)-DNA analyses were all the particles in the DNA bound form and all tetramers. Did this sample have any octameric, non-DNA bound particles?

Supplementary figure 6 shows 2D classification results for all datasets. The particle populations were quite homogeneous in all three cases, however in the 2D alignments of the dataset that yielded the dATP/ATP-loaded NrdR octamer structure a small fraction of the particles was tetrameric (Supplementary Fig. 6B). The relative count of these particles was very low compared to the octameric particles. The NrdR-ATP-dATP-DNA sample contained

DNA in excess (~2.5 μM tetrameric NrdR and 5.5 μM DNA). This could be the reason that NrdR octamers free from DNA weren't detected.

7. Their NrdR(dATP-ATP) structure is octameric and is not compatible with DNA binding. The protein needs to dissociate to a tetramer to bind DNA. How is this accomplished? Is the equilibrium driven to a tetramer in the presence of DNA? The authors should discuss/address this question.

NrdR is present in a tetramer-octamer equilibrium (GEMMA results). Therefore, when NrdR binds to the DNA, more NrdR tetramers are produced. Conceivably, the NrdR tetramer has a higher affinity towards DNA than towards another tetramer.

Along the question above- what are the buried surface areas in the various interfaces? We have performed an analysis of the buried solvent accessible area in the octamer and dodecamer and added this to the relevant sections describing the structures of these oligomers. The results suggest that the dodecamer may be a more stable species in solution. Please see the response to Reviewer 1 question 3 for details.

8. As the NrdR(dATP-ATP)-DNA complex is tetrameric what happens to the other two DNA binding domains within the tetramer? Do they contact any part of the DNA? If not, is the tetramer actually needed for DNA binding or could a dimer bind?

One "dimeric part" of the tetramer binds to NrdR box 1 and the other "dimeric part" of the tetramer binds to NrdR box 2, so the entire tetramer is needed for binding to the promoter sequence.

9. It would seem that dATP + ADP complex would bind DNA with higher affinity as this complex seems to be present as primarily the tetramer (Fig. 4). Can the authors comment on this.

There is no obvious correlation between the frequency of tetrameric particles and binding constants. The binding constant is defined by the interaction between protein and DNA, and in the case of the dATP/ATP-loaded NrdR also the equilibrium between tetramers and octamers.

10. Figure 2 is a key figure. But I found Figure 2A unclear in showing why the octamer cannot bind DNA. Also from the DNA bound complex it is unclear how the tetrameric subunits are associated. It would be helpful if the authors could provide clearer figures here even if it means including density maps and ribbon diagrams of each complex separately. Supplementary videos of the oligomeric assemblies attempt to clarify the arrangements of the monomers in each structure. Comparing the videos of the octameric and tetrameric NrdR structures shows that the DNA-binding interface is not solvent-accessible in the octameric structure. In other words, the DNA is located precisely where the two tetramers in the octamer interact, as shown in figure 2A. Supplementary figure 4A also attempts to illustrate the arrangement of the monomers in all structures in a clearer way, in model-surface view instead of showing the cryo-EM maps.

REVIEWERS' COMMENTS

Reviewer #1 (Remarks to the Author):

In the revised manuscript, the authors have added a few figure illustrations and videos that demonstrate data quality and explain important structural details. The authors also supplemented mutant data to confirm the protein-DNA interaction. It is interesting that the NrdR box does not overlap with the core promoter elements (-35/-10) of *nrdRJ* promoter, but NrdR still represses the promoter activity. The sharp bending of NrdR box caused by NrdR tetramer binding explains well its repressive activity. The authors have addressed all my previous concerns. I would like to support publication of this study in Nature Communications after addressing the following two minor points if other reviewers do not identify other major issues.

1. In supplementary Table 1, the B-factor rows for protein and DNA were switched.
2. The choice of significant digits for the KD values are not consistent.

Reviewer #2 (Remarks to the Author):

Overall, the authors put a lot of effort into the manuscript revision and performed additional mutation experiments, which have addressed all my concerns. Given that the new structures offer interesting insights into the mechanism of different oligomer states of NrdR mediated by ATP or/and dATP, and how NrdR binds to DNA and represses the transcription of *RNR*, this work will be of interest to a sizable audience and is suitable for publication in this journal in the opinion of this reviewer.

Reviewer #3 (Remarks to the Author):

This revised version of the manuscript by Grinberg *et al.*, Logan Sjoberg and Stenmark clarifies many of the comments made by this reviewer and the other reviewers.

There are still a few minor points that the authors should address.

(1) Figure 1A shows a zinc atom in the zinc-ribbon. Yet the authors state they are not comfortable with showing density for these atoms. They should add some wording in the figure legend about this, e.g., this is the most probable location of the zinc atom or some such wording.

(2) The authors do not see perhaps the anticipated data when residue D15 is substituted by alanine. Is it simply that D15 is used to buttress R17 so that the latter residue is positioned to recognize the *indr* boxes specifically? Buttressing interactions of this type are seen frequently in the "wings" of winged HTH DNA binding proteins, which enable minor groove binding typically by an arginine. This is just for the authors to consider.

(3) It is still not clear to this reviewer if the authors tried to fit their ITC data to a two-site model. This should be made clear in the methods section if they did try this or why they did not attempt such a fit.

Reviewer #4 (Remarks to the Author):

The authors have satisfactorily addressed most of the concerns of the reviewers. This is an important advancement for the field and I support publication.

Response to reviewers

Reviewer #1 (Remarks to the Author):

In the revised manuscript, the authors have added a few figure illustrations and videos that demonstrate data quality and explain important structural details. The authors also supplemented mutant data to confirm the protein-DNA interaction. It is interesting that the NrdR box does not overlap with the core promoter elements (-35/-10) of nrdRJ promoter, but NrdR still represses the promote activity. The sharp bending of NrdR box caused by NrdR tetramer binding explains well its repressive activity. The authors have addressed all my previous concerns. I would like to support publication of this study in Nature Communications after addressing the following two minor points if other reviewers do not identify other major issues.

1. In supplementary Table 1, the B-factor rows for protein and DNA were switched.
2. The choice of significant digits for the KD values are not consistent.

1. The incorrect line break in Supplementary Table 1 has been corrected.
2. The Kd digits in figure 4 have been changed to consistency.

Reviewer #2 (Remarks to the Author):

Overall, the authors put a lot of effort into the manuscript revision and performed additional mutation experiments, which have addressed all my concerns. Given that the new structures offer interesting insights into the mechanism of different oligomer states of NrdR mediated by ATP or/and dATP, and how NrdR binds to DNA and represses the transcription of RNR, this work will be of interest to a sizable audience and is suitable for publication in this journal in the opinion of this reviewer.

We thank the reviewer for the positive response and encouraging comments.

Reviewer #3 (Remarks to the Author):

This revised version of the manuscript by Grinberg *et al.*, Logan Sjoberg and Stenmark clarifies many of the comments made by this reviewer and the other reviewers.

There are still a few minor points that the authors should address.

(1) Figure 1A shows a zinc atom in the zinc-ribbon. Yet the authors state they are not comfortable with showing density for these atoms. They should add some wording in the figure legend about this, e.g., this is the most probable location of the zinc atom or some such wording.

As suggested, such wording has been added to the legend of figure 1a.

(2) The authors do not see perhaps the anticipated data when residue D15 is

substituted by alanine. Is it simply that D15 is used to buttress R17 so that the latter residue is positioned to recognize the *ndr* boxes **specifically**? Buttrussing interactions of this type are seen frequently in the "wings" of winged HTH DNA binding proteins, which enable minor groove binding typically by an arginine. This is just for the authors to consider.

We thank the reviewer for this additional suggestion.

(3) It is still not clear to this reviewer if the authors tried to fit their ITC data to a two-site model. This should be made clear in the methods section if they did try this or why they did not attempt such a fit.

We have added in the Materials section, that our attempts to evaluate a two sites model were unsuccessful ("Attempts to fit the data to a two sites model gave too high uncertainty.")

Reviewer #4 (Remarks to the Author):

The authors have satisfactorily addressed most of the concerns of the reviewers. This is an important advancement for the field and I support publication.

We thank the reviewer for the positive response and encouraging comments.